# CEP290 is essential for the initiation of ciliary transition zone assembly

Zhimao Wu[1,2,3], Nan Pang[4], Yingying Zhang[1,2,3], Huicheng Chen[1,2,3], Ying Peng[5], Jingyan Fu[4], Qing Wei[3]*

1 Chinese Academy of Sciences Key Laboratory of Insect Developmental and Evolutionary Biology, Chinese Academy of Sciences Center for Excellence in Molecular Plant Sciences, Institute of Plant Physiology and Ecology, Chinese Academy of Sciences, Shanghai, China, 2 University of Chinese Academy of Sciences, Beijing, China, 3 Center for Energy Metabolism and Reproduction, Institute of Biomedicine and Biotechnology, Shenzhen Institutes of Advanced Technology, Chinese Academy of Sciences, Shenzhen, China, 4 State Key Laboratory of Agrobiotechnology, College of Biological Sciences, China Agricultural University, Beijing, China, 5 Institute of Medicine and Pharmaceutical Sciences, Zhengzhou University, Zhengzhou, China

* qing.wei@siat.ac.cn

**Data Availability Statement:** All relevant data are within the paper and its Supporting Information files.

**Funding:** This work was supported by National Natural Science Foundation of China (http://www.

## Abstract

Cilia play critical roles during embryonic development and adult homeostasis. Dysfunction of cilia leads to various human genetic diseases, including many caused by defects in transition zones (TZs), the "gates" of cilia. The evolutionarily conserved TZ component centrosomal protein 290 (CEP290) is the most frequently mutated human ciliopathy gene, but its roles in ciliogenesis are not completely understood. Here, we report that CEP290 plays an essential role in the initiation of TZ assembly in *Drosophila*. Mechanistically, the N-terminus of CEP290 directly recruits DAZ interacting zinc finger protein 1 (DZIP1), which then recruits Chibby (CBY) and Rab8 to promote early ciliary membrane formation. Complete deletion of CEP290 blocks ciliogenesis at the initiation stage of TZ assembly, which can be mimicked by DZIP1 deletion mutants. Remarkably, expression of the N-terminus of CEP290 alone restores the TZ localization of DZIP1 and subsequently ameliorates the defects in TZ assembly initiation in *cep290* mutants. Our results link CEP290 to DZIP1-CBY/Rab8 module and uncover a previously uncharacterized important function of CEP290 in the coordination of early ciliary membrane formation and TZ assembly.

## Introduction

Cilia are microtubule-based organelles that extend from the surface of most eukaryotic cells. These protrusions are important for diverse cellular functions, including cell signaling, sensory perception, cell motility, and extracellular fluid movement [1–4]. In humans, almost every cell has at least 1 cilium. Due to the ubiquitous distribution and important functions of cilia, ciliary dysfunction results in a wide variety of human genetic disorders, collectively termed ciliopathies [5,6].

Cilia arise from the distal ends of basal bodies (BBs) that are derived from mother centrioles. Depending on the cell type, cilia form through 1 of 2 pathways: the intracellular pathway

nsfc.gov.cn/), Grant 31871357, 32070692 and 31671549 to Q.W. The funders had no role in study design, data collection and analysis, decision to publish, or preparation of the manuscript.

**Competing interests:** The authors have declared that no competing interests exist.

**Abbreviations:** 3D-SIM, three-dimensional structured illumination microscopy; APF, after puparium formation; BB, basal body; BBS, Bardet–Biedl syndrome; CDS, coding sequence; CEP290, centrosomal protein 290; Co-IP, coimmunoprecipitation; CV, ciliary vesicle; CBY, Chibby; DA, distal appendage; DILA, Dilatory; DZIP1, DAZ interacting zinc finger protein 1; EHD1, EH domain-containing protein 1; EM, electron microscope; FBF1, Fas binding factor 1; GFP, green fluorescent protein; gRNA, guide RNA; GST, glutathione S-transferase; IFT, intraflagellar transport; JBTS, Joubert syndrome; LCA, Leber Congenital Amaurosis; MKS, Meckel–Gruber syndrome; NPHP, Nephronophthisis; PCV, preciliary vesicle; PLP, pericentrin-like protein; PVDF, polyvinylidene fluoride; RT-PCR, reverse transcription-polymerase chain reaction; SLS, Senior–Loken syndrome; TEM, transmission electron microscopy; TF, transition fiber; TZ, transition zone; UNC, Uncoordinated; WT, wild-type.

and the extracellular pathway [7,8]. Recent studies on mammalian cells have revealed that, in both pathways, ciliogenesis begins with the docking of Myo-Va–associated small vesicles to mother centriole distal appendages (DAs) [9]. These small vesicles, termed preciliary vesicles (PCVs), then fuse into a large ciliary vesicle (CV) in a process mediated by the membrane-shaping proteins EH domain-containing protein 1 (EHD1) and EHD3 [10]. Then, the Rab11--Rabin8-Rab8 signaling cascade mediates the formation of the early ciliary shaft membrane [9–12]. Perhaps at the same time, the centriole cap protein CP110 is removed, followed by the subsequent assembly of the transition zone (TZ) which is the fundamental ciliary base structure for gating the ciliary compartment [13,14], and then the elongation of the axoneme which is mediated by the evolutionarily conserved intraflagellar transport (IFT) machinery [15].

The TZ is characterized by Y-link–like structures under electron microscopy, which connect proximal axonemal microtubules to the ciliary base membrane and may organize the ciliary necklace composed of annular intramembrane particles on the TZ membrane surface. It has been proposed that the TZ functions as a diffusion barrier to gate the ciliary compartment [16–18]. Dozens of proteins have been identified as TZ components from various model organisms [6,14,19]. Elegant genetic studies on *C. elegans* have grouped known TZ proteins into 3 functional modules, namely, Meckel–Gruber syndrome (MKS), Nephronophthisis (NPHP), and CEP290 [20–22]. Although great progress has been made in understanding the molecular composition and hierarchical assembly of TZs, little is known about how TZ assembly is initiated after basal body docking, or about how early ciliary shaft membrane formation is coordinated with TZ assembly.

CEP290, an important TZ component, is evolutionarily conserved from unicellular *Chlamydomonas* to humans [20,21,23–25] and is one of the most intriguing cilia genes. Mutations in *CEP290* cause several ciliopathies ranging from nonsyndromic retinal degeneration, i.e., Leber Congenital Amaurosis (LCA), to syndromic disorders including, Senior–Loken syndrome (SLS), Joubert syndrome (JBTS), MKS, and Bardet–Biedl syndrome (BBS) [26]. Over 100 *CEP290* mutations have been identified in patients, and these mutations cause a wide variety of phenotypes, ranging from isolated blindness to embryonic lethality [26]. The broad spectrum of diseases and phenotypes associated with *CEP290* mutations highlight the multiple and critical roles of CEP290 in cilia. However, the molecular mechanisms of CEP290 in TZ assembly are still not fully understood.

Here, we discover that CEP290 is essential for TZ assembly initiation in *Drosophila*. Complete deletion of CEP290 blocks ciliogenesis at the stage of TZ assembly initiation in both ciliated sensory neurons and spermatocytes, 2 main ciliated cell types of *Drosophila*. Further studies reveal that the N-terminus of CEP290 directly interacts with DAZ interacting zinc finger protein 1 (DZIP1) and recruits DZIP1 to the TZ. Interestingly, deletion of DZIP1 results in a phenotype similar to *cep290* deletion mutants. Furthermore, we demonstrate that DZIP1 functions upstream of CBY and Rab8 to mediate ciliary membrane assembly. We propose that ciliary membrane assembly is a prerequisite for TZ assembly and that CEP290 coordinates the formation of early ciliary membranes with the assembly of the TZ.

## Results

### Both N-terminal and C-terminal truncation mutants of CEP290 localize to the TZ in *Drosophila*

To determine the domain that mediates the TZ localization of *Drosophila* CEP290, we constructed several green fluorescent protein (GFP)-tagged CEP290 truncation constructs and assessed their subcellular localization in various types of cilia in *Drosophila*: auditory cilia, olfactory cilia, and spermatocyte cilia. It has been proposed that CEP290 bridges axonemal

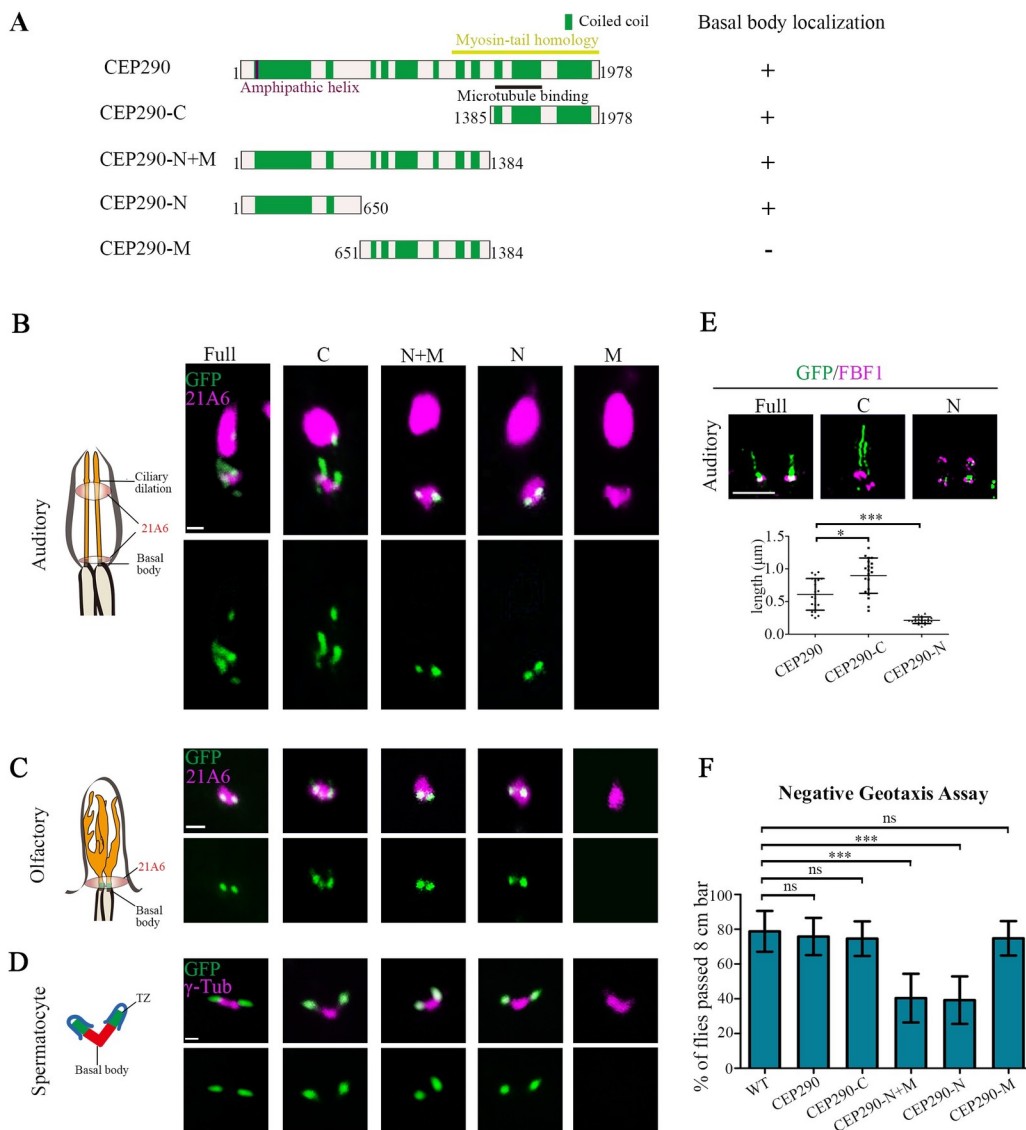

**Fig 1. Both the N-terminal and C-terminal truncation mutants of CEP290 localized to TZs in *Drosophila*.** (A) Schematic representations of CEP290 truncations used in (B–E). (B) Localization patterns of various GFP-tagged CEP290 truncations in auditory cilia. 21A6 marks the ciliary base (red). Both CEP290-N and CEP290-C localized to the ciliary base, but CEP290-M did not. Bar, 1 μm. (C) The localization of various GFP-tagged CEP290 truncations in olfactory cilia. 21A6 marks the ciliary base (red). Both CEP290-N and CEP290-C localized to the ciliary base, but CEP290-M did not. Bar, 1 μm. (D) The localization of various GFP-tagged CEP290 truncations in spermatocytes. γ-Tubulin was used to label the BBs (red). Both CEP290-N and CEP290-C localized to the ciliary base, but CEP290-M did not. Bar, 1 μm. (E) 3D-SIM images of the localization of CEP290, CEP290-N, and CEP290-C in auditory cilia (upper panel), and the corresponding quantification of signal length (lower panel). FBF1 was used to indicate the ciliary base. *n* = 20. Bar, 1 μm. (F) Analysis of the moving behavior of flies overexpressing of CEP290 truncations. Climbing assay was used to assess the fly motility. Motility of flies with overexpression of CEP290-N and CEP290-N+M were significantly compromised. Uncropped images for panels B and C can be found in S1 Raw Image. Numerical data for panel E and F can be found in the file S1 Data. 3D-SIM, three-dimensional structured illumination microscopy; BB, basal body; CEP290, centrosomal protein 290; FBF1, Fas binding factor 1; GFP, green fluorescent protein; ns, not significant; WT, wild-type.

microtubules and ciliary membranes, where its C-terminus contains a microtubule binding domain and N-terminus possesses a highly conserved membrane-binding amphipathic α-helix motif (Fig 1A) [23,27,28]. Consistent with these findings, the C-terminal fragment of

CEP290 containing the microtubule binding domain (CEP290-C, from aa 1385 to the end) localized to the TZ in all examined cilia in *Drosophila* (Fig 1A–1E), and the N-terminal fragment of CEP290 (CEP290-N, comprising aa 1–650) alone was also located at the TZ, but the middle fragment (CEP290-M, aa 651–1384) was not (Fig 1A–1E). The fact that CEP290-N alone targets to the TZ suggests that the N-terminus of CEP290 has a TZ target signal independent of the C-terminal microtubule binding domain. Notably, wild-type (WT) flies with overexpression of the N-terminus of CEP290 showed defective motility (Fig 1F), a phenotype commonly associated with cilia dysfunction.

The length of the TZ in different types of cilia is variable in *Drosophila*. TZs of auditory cilia are significantly longer than those of olfactory cilia and spermatocyte cilia [27]. TZ in auditory cilia contains 2 longitudinal subdomains: The proximal one comprises all TZ components, whereas the distal one comprises CEP290 only [27]. We did see that the CEP290 signal in auditory cilia was longer than that in other cilia (Fig 1B). However, we observed that the distribution of CEP290-N signal was significantly more restricted than the full-length CEP290 signal (Fig 1B), suggesting that CEP290-N is restricted toward the proximal part of the TZ. On the other hand, we observed that the signal of CEP290-C was significantly longer than that of CEP290 (Fig 1B), which was also seen in olfactory cilia (Fig 1C). Different distributions of CEP290, CEP290-N, and CEP290-C in TZs of auditory cilia were further confirmed by three-dimensional structured illumination microscopy (3D-SIM) (Fig 1E). These results suggest that the N-terminus of CEP290 might have an inhibitory effect on the localization of its C-terminus.

## CEP290 is essential for the initiation of TZ assembly in spermatocytes

The fact that a short N-terminal truncation mutant of CEP290 was targeted to the TZ suggests that the previously reported *cep290* mutant (*cep290^mecH*, lacking the C-terminal sequence from aa 1471 to the end) may be not a null mutant [24]. To better understand the function of CEP290, especially its N-terminus, we employed CRISPR-mediated genome engineering and generated a putative null mutant, *cep290^1* (c.469-1174Del), in which site-specific deletion leads to a frameshift, leaving only 156 aa of the N-terminus (Fig 2A and S1A Fig). In order to compare the phenotype differences between the *cep290^1* mutants and C-terminus deletion mutants, we additionally generated *cep290^ΔC* (c.4156delG), in which a deletion results in a frameshift and C-terminus (aa 1386-end) loss (Fig 2A and S1B and S1C Fig). As expected, immunofluorescence staining of CEP290 in spermatocytes with an antibody against its N-terminal showed that the N-terminus of CEP290 was completely lost in *cep290^1* mutants, but it was still present in *cep290^ΔC* mutants (S1D Fig). Notably, compared with WT, the signal intensity of CEP290 in *cep290^ΔC* mutant was significantly reduced (S1D Fig). This is consistent with the transcript level of *cep290* in *cep290^ΔC* mutant, which was also significantly lower than that of WT (S1B Fig), probably due to nonsense-mediated mRNA decay.

Similar to previously reported alleles [24,29], both *cep290^1* and *cep290^ΔC* were severely uncoordinated during walking and flying, and they had severe defects in both touch sensitivity and hearing sensation (S1E Fig), typical phenotypes of cilia deficiency mutants. Importantly, all these defects could be rescued by expression of *UAS-cep290* under the pan-neural driver *elav-GAL4* (S1E Fig).

First, we focused on cilia in spermatocytes. *Drosophila* spermatocyte ciliogenesis has several unique features (Fig 2B): (1) Cilia grow from both mother and daughter centrioles [30]; (2) cilia have only a TZ region, with no axoneme above the TZ [31]; (3) cilia are assembled on the cell surface in the early G2 phase but are internalized into the cytosol along with BBs in later stages, as BBs need to be used to assemble the spindle for meiosis [32,33]; and (4) during

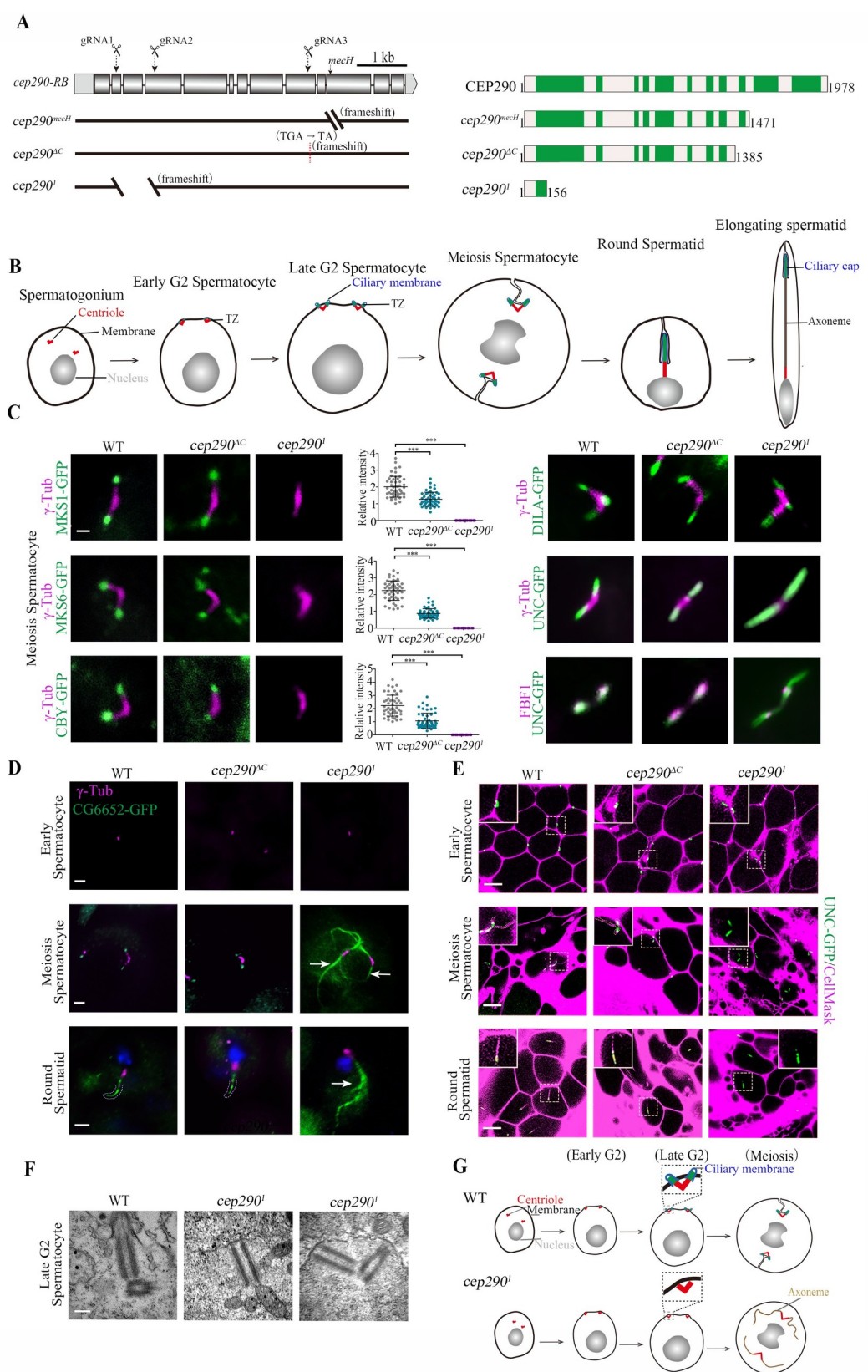

**Fig 2. CEP290 is essential for TZ assembly initiation in spermatocytes.** (A) Generation of *cep290* knockout flies. Schematics show the genomic (left) and protein (right) structures of CEP290 and its mutants. The gRNA target sites are shown. *cep290^{ΔC}* (c.4156delG) lacks the C-terminal region from aa 1386 to the end, and *cep290^{1}* has a deletion in cDNA from nt 469 to 1,174, resulting in a frameshift and leaving only 156 aa. (B) Schematic diagram of dynamic changes in cilia/TZs during different stages of spermatogenesis. Cilia grow from both mother and daughter centrioles in *Drosophila* spermatocytes. V-shaped centriole pair docks to the plasma membrane in early G2 phase, and then assemble cilia/TZs. During meiosis, primary cilia are internalized along with BB, as BB needs to be used to assemble spindle for meiosis. During the process of internalization, cilia and the plasma membrane remain connected. After meiosis, spermatids elongate and mature. At this stage, cilia, also known as ciliary caps of spermatids, are extended and move away from the BB. (C) Representative images of the localization of TZ-related proteins in WT, *cep290^{ΔC}* mutants, and the *cep290^{1}* mutants. Compared with WT flies, the signal of MKS1-GFP, MKS6-GFP, and CBY-GFP was significantly reduced in *cep290^{ΔC}* mutants but completely lost in *cep290^{1}* mutants. The corresponding quantitative relative fluorescence intensities were shown on the right. Numerical data can be found in the file S1 Data. *n* = 50 centrioles over 5 flies. Targeting of DILA and UNC was not affected in *cep290^{ΔC}* mutants and the *cep290^{1}* mutants, but UNC-GFP abnormally extended above the TF in *cep290^{1}* mutants. Bar, 1 μm. (D) Representative images of the localization of CG6652-GFP in spermatocytes and spermatids in the indicated genetic background. CG6652 marks the primary ciliary axoneme and extends abnormally (arrows) in *cep290^{1}*. γ-Tubulin (red) was used to label the centrosomes. Bars, 2 μm. (E) Live imaging of BB docking in WT flies, *cep290^{ΔC}* mutants, and *cep290^{1}* mutants. The PM was labeled with CellMask, and the BBs were indicated by UNC-GFP. In *cep290^{ΔC}* mutant, no defects were observed in the connection between the BBs and the PM. In *cep290^{1}* mutants, the BBs could initially migrate to the PM, but the connection between the BBs and the membrane disassociated in later stages. Bars, 10 μm. (F) EM images showing that BBs are close to the membrane in *cep290^{1}* mutants but that no ciliary bud is observed. Bar = 200 nm. (G) Schematic diagrams of BBs and cilia in different stages of spermatocytes in WT flies and *cep290^{1}* mutants. The centriole pair migrates and docks to the PM in early spermatocytes, and then the TZ is assembled, which stabilizes the connection between the PM and BBs. In *cep290^{1}* mutant, the centriole pair initially migrates to the plasma membrane, but the TZ fails to form, leading to disassociation of the BB from the PM in later stages, and causing abnormal elongation of the axoneme. BB, basal body; CEP290, centrosomal protein 290; DILA, Dilatory; gRNA, guide RNA; PM, plasma membrane; TZ, transition zone; WT, wild-type.

spermatid elongation after meiosis, cilia (also known as ciliary caps at this stage) extend and move away from BBs [24].

To analyze cilia/TZs formation in spermatocytes, we examined the localization of TZ core proteins (MKS1 and MKS6) and TZ-related proteins (CBY and DILA/CEP131). CBY and Dilatory (DILA) cooperate to build the TZ in *Drosophila* [34], but their localization in the TZ is very different; CBY completely colocalizes with MKS1 and MKS6 near the TZ membrane in spermatocytes, whereas DILA is inside the lumen of the TZ axoneme. Consistent with previous reports [24], we observed that all these proteins were present at the tips of BBs in *cep290^{ΔC}* mutant, although the signal intensity of MKS1, MKS6, and CBY was significantly reduced (Fig 2C). However, surprisingly, we observed that MKS1, MKS6, and CBY were completely lost from the tips of BBs in *cep290^{1}* mutants (Fig 2C), while the distribution of DILA was normal. These results indicate that CEP290 is essential for TZ assembly.

Strikingly, we observed that, compared with that in WT flies and *cep290^{ΔC}* mutants, the signal of Uncoordinated (UNC, the homolog of mammalian OFD1) was aberrantly elongated in *cep290^{1}* mutants (Fig 2C). Colabeling of UNC with the transition fiber (TF) protein Fas binding factor 1 (FBF1) indicated that the elongation occurred above the TF (Fig 2C), suggesting that the ciliary axoneme, rather than the centriole, was elongated. To confirm this, we labeled primary cilia axonemes with CG6652-GFP, which specifically labels axoneme in cilia of spermatocytes and ciliary caps of round spermatids [34]. Indeed, as shown in Fig 2D, axonemes extended abnormally in *cep290^{1}* mutants, but not in WT flies or *cep290^{ΔC}* mutants.

Abnormal spermatocyte ciliary axoneme elongation has previously been observed in *cby; dila* double mutants and has been attributed to defects in basal body docking and the absence of a TZ membrane cap [34]. Accordingly, we observed that the association between BBs (as shown by UNC-GFP) and the plasma membrane (labeled with CellMask, Invitrogen, Carlsbad, California, United States of America) was defective in later spermatocytes and spermatids in *cep290^{1}* mutants, but not in WT flies or *cep290^{ΔC}* mutants (Fig 2E). However, in the early spermatocyte stage, we observed that BBs were still attached/close to the membrane (Fig 2E). To

further confirm this result, we employed transmission electron microscopy (TEM). Consistent with our immunofluorescence observations, BBs were able to migrate to the plasma membrane in *cep290¹* mutants, but no cilia-like structures and TZs formed (Fig 2F). Taken together, we proposed that in *cep290¹* mutants, BBs initially migrate to the plasma membrane, but the TZ assembly initiation is completely blocked, such that primary cilia/TZs never form. During the process of BB internalization in later stages, due to the absence of the ciliary cap, the stable association between the BB and the cell membrane was unable to be maintained, and the BB's microtubules were abnormally elongated within the cytoplasm (Fig 2G).

In addition to the initiation of TZ assembly, spermatogenesis was also very different between *cep290¹* mutants and *cep290^{ΔC}* mutants. As previously reported [24], *cep290^{ΔC}* mutants had mild defects in spermatid development and male fertility (S1F and S1G Fig); however, sperm cysts were severely defective in elongation in *cep290¹* mutants, although the overall sizes of their testes were comparable (S1F Fig). Subsequently, no mature sperms were produced, and males were completely infertile in *cep290¹* mutants (S1G Fig). TEM examination of testes showed that flagellar axonemes were almost completely lost in *cep290¹* mutants (S1H Fig). Similar severe defects in the spermatid axoneme were also observed in TZ absent mutants [34], but the underlying mechanism remains poorly understood.

## CEP290 is critical for the initiation of TZ assembly in sensory neurons

Next, we asked whether CEP290 is also involved in TZ assembly initiation in sensory neurons. Previously, EM data have shown that the auditory cilium in *cep290^{mecH}* mutants lacks the TZ and ciliary axoneme [27]. Consistent with this, complete loss of cilia was observed in *cep290¹* and *cep290^{ΔC}* mutants by both immunofluorescence and electron microscope (EM) assays (Fig 3A and 3B). Our EM data showed that BBs were very close to the plasma membrane, but no apparent TZs formed in either *cep290¹* or *cep290^{ΔC}* mutants (S1I Fig). Although, under EM, we did not find any obvious differences in cilia base structures between *cep290¹* and *cep290^{ΔC}* mutants; we found that, similar to that of spermatocyte cilia, the localization of TZ components in auditory cilia were drastically different in different *cep290* mutants. All TZ proteins examined were present at the BB in *cep290^{ΔC}* mutants, although their signal intensities were severely reduced compared to WT. However, remarkably, these signals were completely lost in *cep290¹* mutants (Fig 3C). Similar results were also observed in olfactory cilia (Fig 3C). These results suggest that in sensory neurons, the CEP290 N-terminus may be essential for initial TZ assembly, whereas TZ elongation or maturation may involve a tissue-specific requirement for the C-terminus of CEP290.

Taken together, our results indicate that CEP290 is essential for the initiation of TZ assembly in both sensory neurons and spermatocytes. Of note, without a robust antibody to survey expression to validate levels of CEP290 protein expression within our alleles, we cannot fully unpick how much of our phenotypes are due solely to reduced levels versus requirement for specific domains. We suspect it is likely a combination of both.

## DZIP1 is a TZ protein interacting with the N-terminus of CEP290

To elucidate the role of CEP290 in TZ assembly initiation, we employed a yeast two-hybrid assay to search for TZ-related proteins that interact with the N-terminus of CEP290 and identified DZIP1. *Drosophila* DZIP1 is the ortholog of mammalian DZIP1 and DZIP1L (S2A–S2D Fig), both have been implicated in ciliogenesis [35–38]. In our yeast two-hybrid assay, DZIP1 interacted with the N-terminus of CEP290, but did not interact with its C-terminus (Fig 4A). The interaction was further confirmed by the glutathione S-transferase (GST) pull-down assay (Fig 4B and S3A Fig). Both the N-terminus and C-terminus of DZIP1 interacted with the

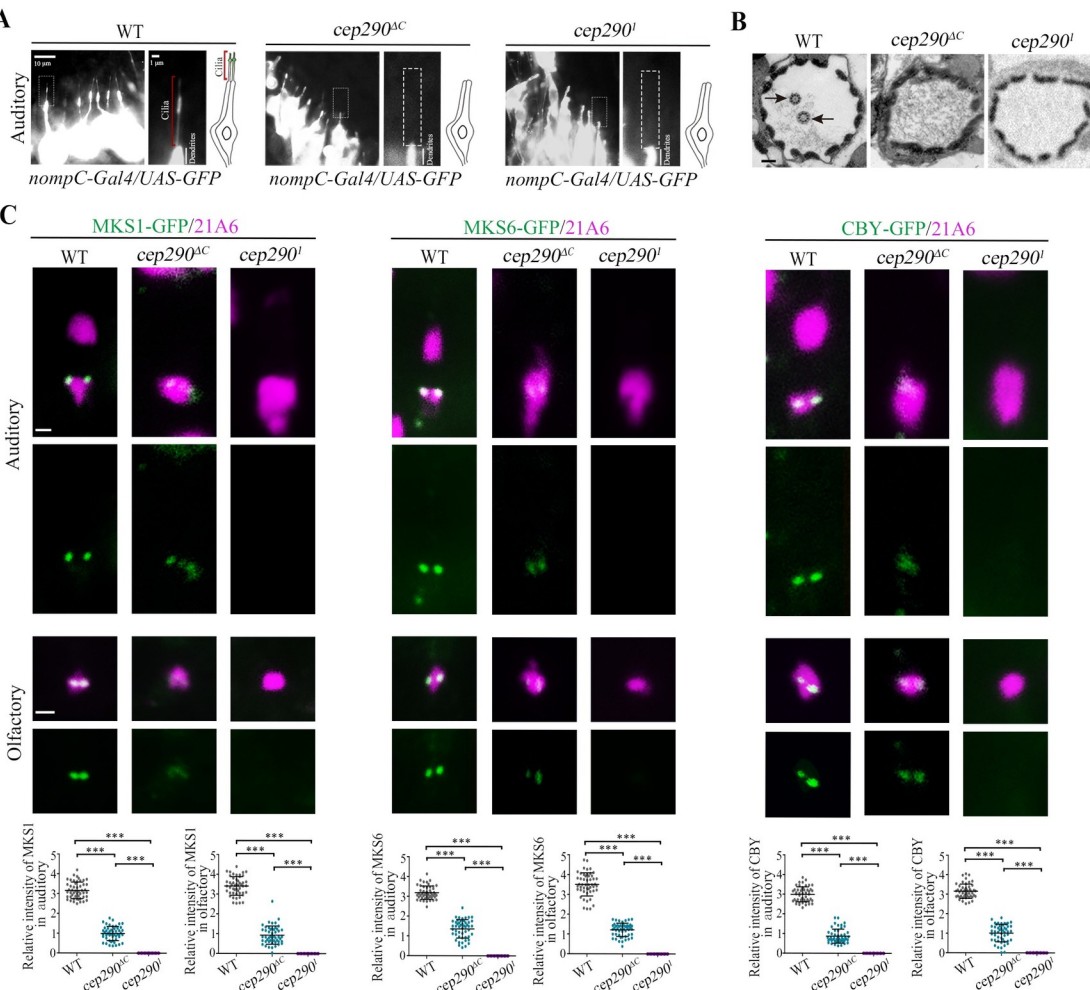

**Fig 3. CEP290 is critical for TZ assembly initiation in sensory neurons.** (A) Cilium morphology in auditory cilia as labeled by *nompC-Gal4; UAS-GFP* in WT flies and *cep290¹* mutants. Cilia were completely lost in *cep290¹* mutants. (B) EM analysis of cross sections of auditory cilia. In WT flies, each scolopidium possessed 2 ciliary axonemes (arrows), but no ciliary axonemes were observed in the scolopidia of *cep290* mutants. Bar, 200 nm. (C) Localization of various TZ proteins in auditory cilia and olfactory cilia in WT flies, *cep290^{ΔC}* mutants, and *cep290¹* mutants, and corresponding relative fluorescence intensity quantitation. In *cep290^{ΔC}* mutant, TZ proteins, MKS1-GFP, MKS6-GFP, and CBY-GFP were still present at the ciliary base, although the signals were severely reduced. However, these signals were completely lost in *cep290¹* mutants. 21A6 (red) was used to label the ciliary base. Uncropped images for panel C can be found in S1 Raw Image. Numerical data for panel C can be found in the file S1 Data. The bars and error bars represent the means and SDs, respectively. *n* = 50 basal bodies over 5 flies. Scale bars, 1 μm. CEP290, centrosomal protein 290; EM, electron microscope; SD, standard deviation; TZ, transition zone; WT, wild-type.

CEP290 N-terminus. DZIP1-N (aa 1–293) interacted with both CEP290 N (aa 401–650) and CEP290 N (aa 651–887); DZIP1-C (aa 294–737) interacted with CEP290 N (aa 651–887) (S3B Fig). To further confirm the interaction, we expressed GFP-tagged CEP290 and HA-tagged DZIP1 in cultured *Drosophila* S2 cells and performed coimmunoprecipitation (Co-IP) using GFP-trap beads. Western blots of the GFP-pulldowns confirmed the binding between CEP290 and DZIP1 (Fig 4C).

While the role of *Drosophila* DZIP1 in ciliogenesis was unknown when we started this project, a recent study showed that DZIP1 was localized to the TZ and was required for ciliogenesis in *Drosophila* [29]. Consistent with this report, we observed that *Drosophila* DZIP1 localized to the basal body in all cilia examined (S4A and S4B Fig). In spermatocytes, DZIP1 was found

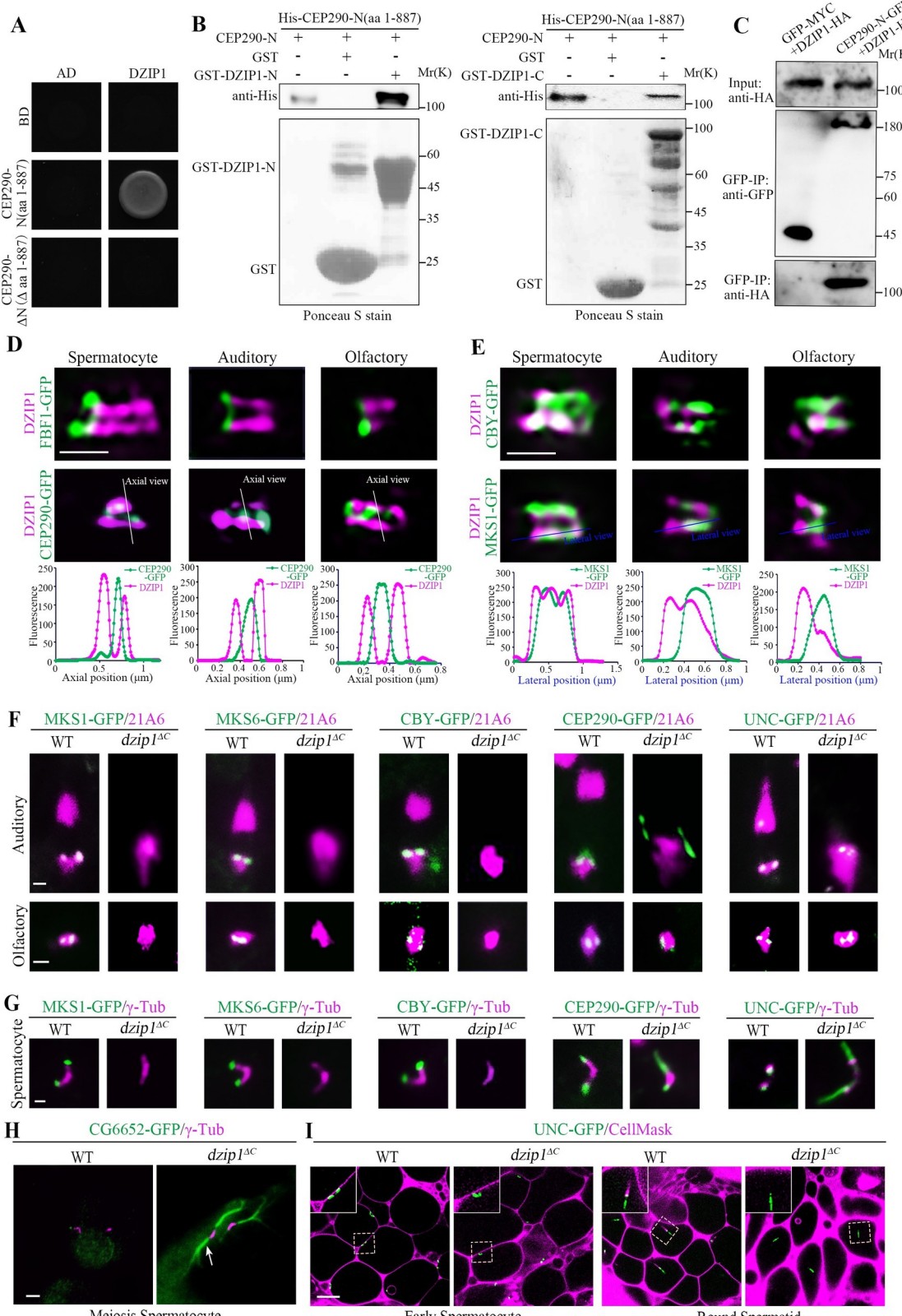

**Fig 4. DZIP1 interacts with the N-terminus of CEP290, and DZIP1 deletion mutants mimic the phenotype of *cep290^1* mutants.**
(A) Yeast two-hybrid assay results showing that DZIP1 interacts with CEP290-N (aa 1–887), but not with CEP290-ΔN (Δ aa 1–887).

(B) GST pull-down assay confirmed the interaction between CEP290-N and DZIP1 in vitro. Both DZIP1-N (aa 1–293) and DZIP1-C (aa 294–737) interacted with CEP290-N. Upper panel, blotting with anti-His antibodies. Lower panel, Ponceau S staining of GST, GST-DZIP1-N, or GST-DZIP1-C. (C) CEP290-N and DZIP1 were coimmunoprecipitated in S2 cells. S2 cells were transiently transfected with CEP290-N-GFP and DZIP1-HA; 48 h later, cells were subjected to Co-IP using GFP-trap beads. (D, E) 3D-SIM images of the localization of DZIP1 on TZs in different types of cilia. (D) In all cilia, DZIP1 localized distal to the TF protein FBF1 and surrounded the C-terminal GFP-tagged CEP290. (E) In spermatocytes, DZIP1 overlapped with MKS1 and CBY. In sensory neurons, DZIP1 localized above FBF1, below MKS1, and partially overlapped with MKS1. Bars, 500 nm. (F) The TZ proteins MKS1-GFP, MKS6-GFP, and CBY-GFP were completely lost from the ciliary base in sensory neurons in $dzip1^{\Delta C}$ mutants. CEP290-GFP and UNC-GFP were still present at the ciliary base. Notably, the signal of CEP290 was abnormally expanded in auditory cilia. 21A6 (red) marks the ciliary base. Bar, 1 μm. (G) MKS1, MKS6, and CBY were almost completely lost in spermatocytes of $dzip1^{\Delta C}$ mutants. CEP290 and UNC were still present, but the signals were longer in mutants than in WT flies. γ-Tubulin was used to label the BBs (red). Bar, 1 μm. (H) Representative images of aberrantly elongated and branched spermatocyte ciliary axonemes (white arrow) labeled by CG6652-GFP in $dzip1^{\Delta C}$ mutants. γ-Tubulin was used to label the BBs (red). Bar, 2 μm. (I) Live imaging of the connection between BBs and the PM in WT flies and $dzip1^{\Delta C}$ mutants. The membrane was labeled with CellMask, and the BBs were indicated by UNC-GFP. In $dzip1^{\Delta C}$ mutants, the BBs migrated to the PM in the early spermatocyte stage but failed to maintain the cilium-PM connection in subsequent processes. Bar, 10 μm. Numerical data for panels D and E can be found in the file S1 Data. Uncropped images for panel F can be found in S1 Raw Image. Uncropped immunoblots for panels B and C can be found in S2 Raw Image. 3D-SIM, three-dimensional structured illumination microscopy; BB, basal body; CBY, Chibby; CEP290, centrosomal protein 290; Co-IP, coimmunoprecipitation; DZIP1, DAZ interacting zinc finger protein 1; FBF1, Fas binding factor 1; GFP, green fluorescent protein; GST, glutathione S-transferase; HA, the hemagglutinin (HA) tag; MKS1, Meckel syndrome type 1; MKS6, Meckel syndrome type 6; PM, plasma membrane; TZ, transition zone; UNC, Uncoordinated; WT, wild-type.

to localize to the tips of BBs. In spermatids, like TZ proteins, DZIP1 migrated along with the ciliary cap to the flagellar tip. To more accurately determine the subcellular localization of DZIP1, we employed 3D-SIM and observed that in all types of cilia, DZIP1 localized above the TF protein FBF1, surrounded the C-terminal GFP-tagged CEP290 which labels the center of the TZ [27] (Fig 4D). In spermatocytes, DZIP1 overlapped completely with the TZ protein MKS1 and CBY along TZ membranes (Fig 4E). In spermatids, DZIP1 was enriched at the ring centriole like CBY, but did not extend distally along the whole ciliary cap like MKS1 and MKS6 (S4C Fig). In sensory neurons, there was an obvious gap between the TF protein FBF1 and the TZ component MKS1 (S4D Fig). We observed that DZIP1 was above FBF1, but clearly below MKS1 and partially overlapped with MKS1, indicating that DZIP1 occupied the gap region between FBF1 and MKS1, and further extended distally, partially colocalizing with MKS1 (Fig 4E). Therefore, there are tissue-specific differences in DZIP1-related ciliary base structures. Overall, *Drosophila* DZIP1 is a TZ protein interacting with the N-terminus of CEP290.

## *dzip1* deletion mutants mimic the phenotype of *cep290* mutants

To investigate the functional relationship between CEP290 and DZIP1, we generated a deletion mutant of $dzip1^{\Delta C}$ (c.1069-1529Del), in which the C-terminus (aa 357-end) of DZIP1 is lost (S5A–S5C Fig). Consistent with the recent report [29], *dzip1* mutant exhibited typical cilia mutant phenotypes. The flies were completely uncoordinated, unable even to stand, and were severely deficient in touch and hearing (S5D Fig). Introducing a WT *DZIP1 cDNA* transgene into the mutants fully rescued these deficiencies (S5D Fig), indicating that DZIP1 was the gene responsible for these defects.

Interestingly, we found that our *dzip1* mutants mimicked the phenotype of *cep290*[1] mutants. In sensory cilia of *dzip1* mutants, TZ components MKS1, MKS6, and CBY were completely absent from the TZ, but the recruitment of CEP290 and UNC was not affected (Fig 4F). And cilia were almost completely lost in auditory cilia of $dzip1^{\Delta C}$ mutant as demonstrated by both immunofluorescence and EM assays (S5E and S5F Fig). In spermatocytes of *dzip1* mutants, MKS1, MKS6, and CBY were almost completely lost but CEP290 persisted (Fig 4G). Furthermore, similar to *cep290*[1] mutants, ciliary axonemes extended aberrantly in *dzip1*

mutants, as shown by UNC-GFP and CG6652-GFP (Fig 4G and 4H). Additionally, deletion of DZIP1 resulted in the separation of the BBs and the plasma membrane in late spermatocytes (Fig 4I). All these observations are consistent with the recent report [29], indicating that like CEP290, DZIP1 is essential for TZ assembly initiation.

## CEP290 recruits DZIP1 to regulate TZ assembly

*cep290¹* and *dzip1* mutants have similar phenotypes, and DZIP1 is dispensable for the recruitment of CEP290 (Fig 4F and 4G). Interestingly, the DZIP1 signal was completely lost in both spermatocyte cilia and sensory neuron cilia in *cep290¹* mutants, but it is still existed in the *unc*, *dila*, and *cby* mutants (Fig 5A and 5B). It has been reported that CEP290 is lost in *cby;dila* double mutants [34]; accordingly, no DZIP1 was observed in this double mutants (Fig 5A and 5B). Notably, in *cep290^{ΔC}* mutants, DZIP1 was present in all types of cilia, suggesting that the remaining N-terminus of CEP290 has a role in recruiting DZIP1 to the TZ. However, compared with WT, the signal intensity of DZIP1 decreased significantly in sensory cilia in *cep290^{ΔC}* mutants, and it extended abnormally into the whole cilia/TZs in spermatids (Fig 5B and 5C), suggesting that the CEP290 C-terminus may have tissue-specific roles in regulating the localization of DZIP1.

Because the N-terminus of CEP290 directly interacts with DZIP1 (Fig 4A–4C), we hypothesized that CEP290-N may directly recruit DZIP1 to regulate TZ assembly. If so, expressing the N-terminus of CEP290 should ameliorate the defects in TZ assembly initiation in *cep290¹* mutants. To test this hypothesis, we expressed CEP290 truncations in *cep290¹* mutants individually. Interestingly, we found that overexpression of CEP290-N restored the signal of DZIP1 on the TZ in *cep290¹* mutants, but the middle and C-terminal fragments did not (Fig 5D). Remarkably, we observed the return of MKS1 signal in both spermatocytes and spermatids, indicating that overexpression of the N-terminus correspondingly attenuated the defects in TZ formation in *cep290¹* mutants (Fig 5E). Accordingly, the BB docking defects in late spermatocytes were also ameliorated after overexpression of CEP290-N (Fig 5F). Notably, the flies with overexpression of CEP290-N were still uncoordinated; this is because that CEP290 has other roles independent of its N-terminal and DZIP1; CEP290 is not only required for the initiation of ciliogenesis and TZ assembly, but also required for TZ maturation and axoneme formation in sensory neurons.

## DZIP1 regulates TZ assembly upstream of CBY and Rab8

Lapart and colleagues reported that *Drosophila* DZIP1 directly interacts with CBY and function upstream of CBY-FAM92 module to regulate the TZ assembly [29]. However, *dzip1* mutant has much more severely defective phenotypes than *cby* or *fam92* single mutants. Ciliogenesis is completely blocked in our *dzip1* mutants, but it is partially affected in *cby* or *fam92* single mutants [29,39]. These results suggest that defects caused by deletion of DZIP1 cannot be attributed to CBY-FAM92 pathway only, other binding factors of DZIP1 should also be involved.

It has been reported that mammalian DZIP1 interacts with ciliary membrane formation regulator Rab8 [40]. Rab8 is a core regulator of membrane vesicle trafficking whose role in ciliary membrane formation has been extensively studied [9,41]. To determine if DZIP1 interacts with Rab8 in *Drosophila*, we performed the GST pull-down assay. Interestingly, we found that the N-terminus of DZIP1 interacted with Rab8, while the C-terminus of DZIP1 did not (Fig 6A). Consistent with what has been reported in mammals [40], DZIP1 preferentially interacted with the Rab8^{GDP}-mimicking mutant Rab8^{T22N}. To further confirm the interaction, we transiently transfected *Drosophila* S2 cells with DZIP1-GFP and Rab8-HA. GFP-pulldowns from

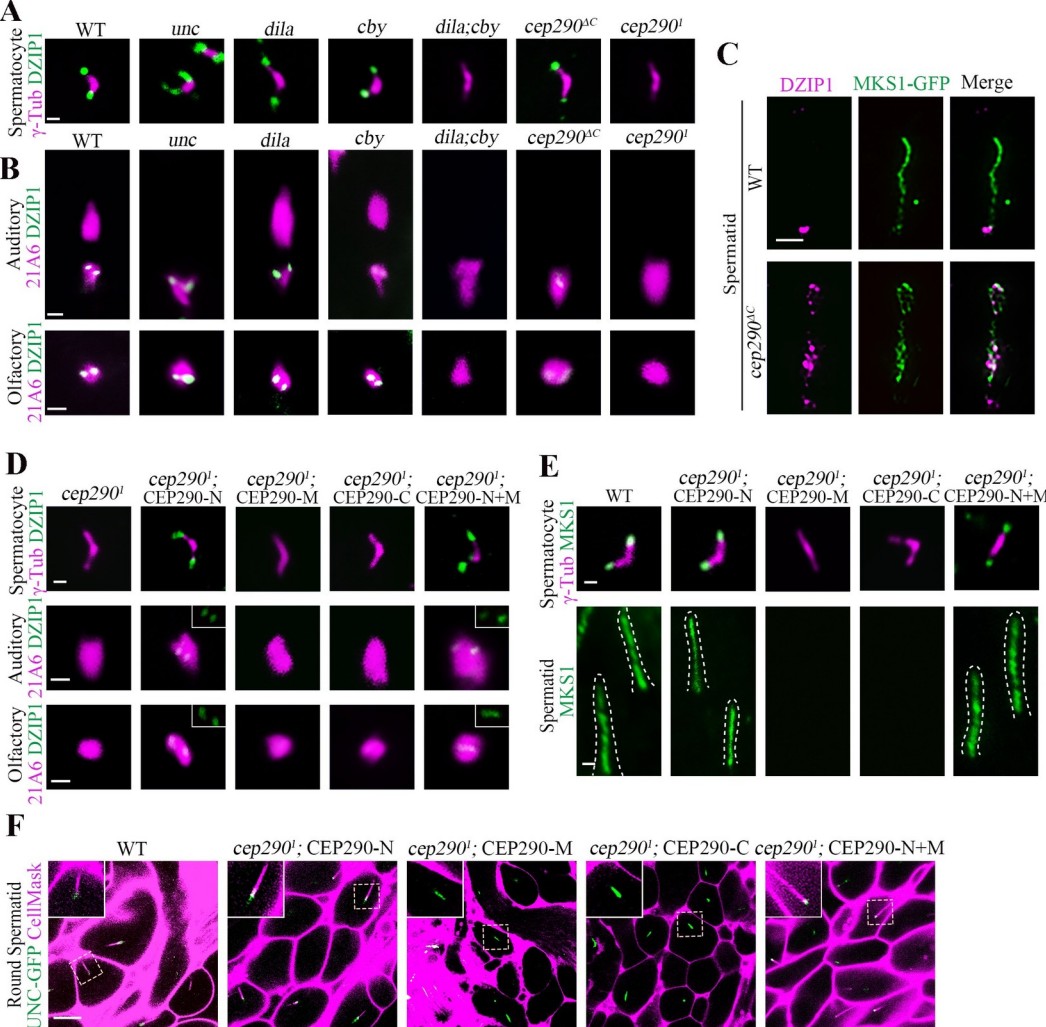

**Fig 5. The N-terminus of CEP290 recruits DZIP1 to regulate TZ assembly.** (A) Localization of DZIP1 in spermatocytes in the indicated genetic background. DZIP1-GFP was completely lost in *cep290¹* mutants and *cby;dila* double mutants, but existed in other mutants, including *cep290^{ΔC}* mutants. γ-Tubulin (red) was used to label the BBs. Bar, 1 μm. (B) Localization of DZIP1 in sensory neurons in the indicated genetic background. In *cep290¹* mutants, DZIP1-GFP was completely lost. In *cep290^{ΔC}* mutants, DZIP1 was present, but the signal was severely reduced in sensory neurons. 21A6 staining indicates the ciliary base in sensory neurons. Uncropped images can be found in S1 Raw image. Bars, 1 μm. (C) The C-terminus of CEP290 is required to limit the DZIP1 signal at the ring centriole in spermatids. Bar, 1 μm. (D) Overexpression of the CEP290 N-terminus (aa 1–650) ameliorated the localization defects of DZIP1 in *cep290¹* mutants. Bars, 1 μm. (E) Overexpression of the CEP290 N-terminus rescued the localization of MKS1 in spermatocytes and spermatids in *cep290¹* mutants. Bars, 1 μm. (F) Overexpression of CEP290 N-terminus restored the connection between the BBs and the membrane in *cep290¹* mutants. CellMask was used to label the membrane, and UNC-GFP indicates the BBs. Bar, 10 μm. BB, basal body; CEP290, centrosomal protein 290; DZIP1, DAZ interacting zinc finger protein 1; MKS1, Meckel syndrome type 1; TZ, transition zone.

cell extracts confirmed that DZIP1 did interact with Rab8 (Fig 6C). Therefore, Rab8 is a binding factor of DZIP1 in *Drosophila*.

The function of Rab8 in *Drosophila* ciliogenesis has not been investigated yet. We observed that Rab8 localized to the ciliary base in both auditory neurons and olfactory neurons. And consistent with what is observed in mammals, the Rab8^{GTP}-mimicking mutant RAB8^{Q67L} was frequently observed inside cilia (S6A and S6B Fig). Interestingly, DZIP1 and CEP290 were

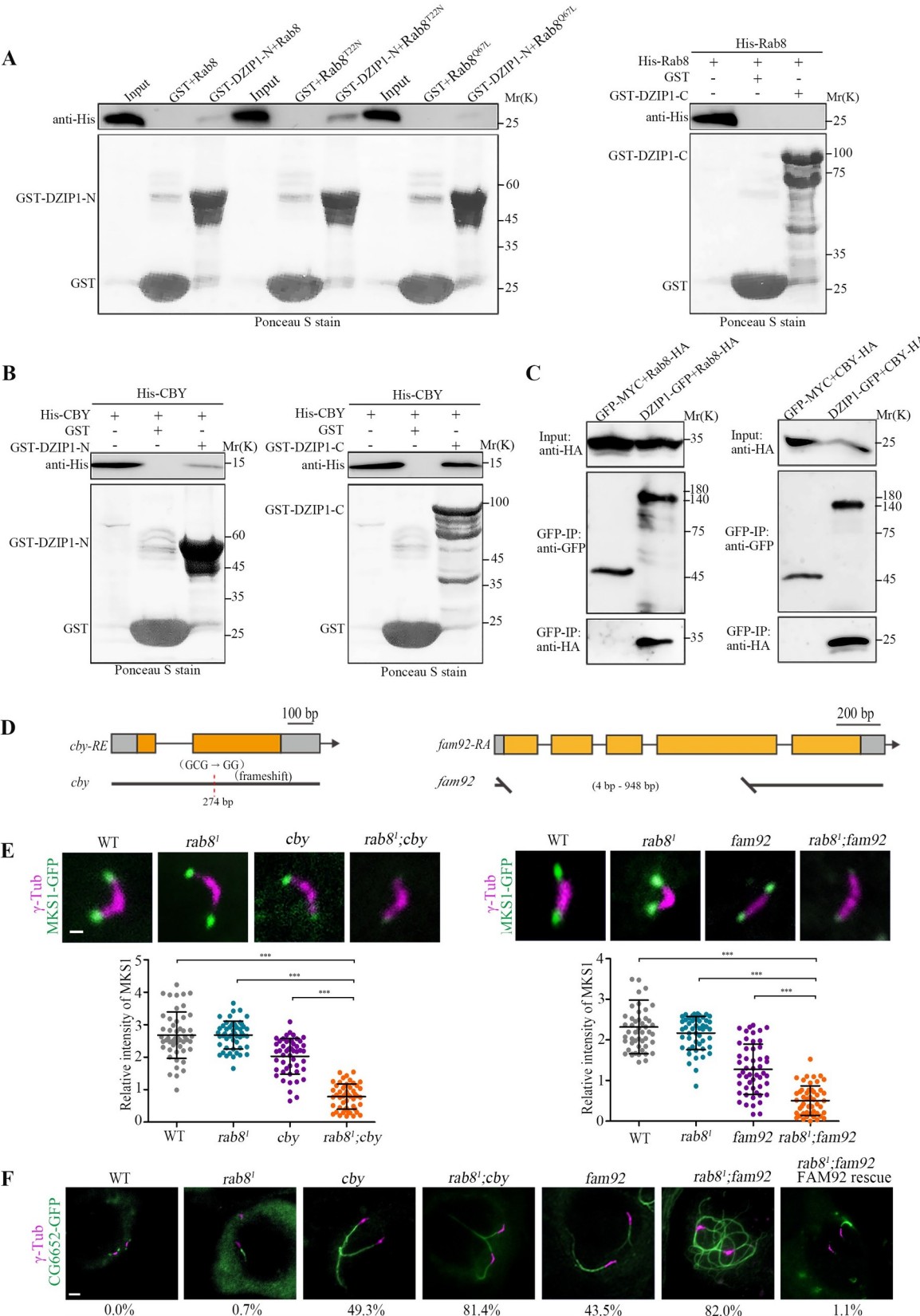

**Fig 6. DZIP1 recruits CBY and Rab8 to promote TZ assembly.** (A) The N-terminus of DZIP1 interacted with Rab8 in a GST pull-down assay, but the C-terminus did not. And DZIP1 preferentially bound to Rab8$^{T22N}$. (B) DZIP1 interacted directly with CBY in a GST pull-down assay. (C) DZIP1-GFP and Rab8-HA, DZIP1-GFP, and CBY-HA were coimmunoprecipitated in cultured *Drosophila* S2 cells. (D) Schematics of *cby* and *fam92*. *cby* (c.121delC) are predicted to encode a truncated protein lacking the C-terminal region from aa 41 to the end. *fam92* (c. 4-789Del) deletes a large portion (262 out of 395 amino acids) of FAM92 protein. (E) Localization of MKS1-GFP in spermatocytes in the indicated genetic background (upper panel) and the corresponding quantitative relative fluorescence intensities (lower panel). Compared with any single mutants, MKS1-GFP was significantly reduced in *rab8$^1$*; *cby* double mutants and in *rab8$^1$*; *fam92* double mutants. The bars and error bars represent the means and SDs, respectively. *n* = 50 centrioles over 5 flies. Scale bars, 1 μm. (F) Images of CG6652-GFP signals in late spermatocytes in WT flies, *rab8$^1$* mutants, *cby* mutants, *fam92* mutants, *rab8$^1$*; *cby* double mutants, and *rab8$^1$*; *fam92* double mutants. Compared to WT, *rab8$^1$*, *cby*, and *fam92*, the percentage of centriole pairs with abnormally elongated CG6652 signal increased significantly in *cby; rab8* double mutant and in *rab8$^1$*; *fam92* double mutants. Overexpression of FAM92 rescued the elongated CG6652 phenotype of *fam92*, *rab8$^1$* double mutants. *n* = 2,000 over 10 flies for *rab8* mutants, *n* = 500 over 5 flies for others. Scale bars, 2 μm. Uncropped immunoblots can be found in S2 Raw Image. Numerical data can be found in the file S1 Data. CBY, Chibby; DZIP1, DAZ interacting zinc finger protein 1; FAM92, Family with sequence similarity 92; GFP, green fluorescent protein; GST, glutathione S-transferase; HA, the hemagglutinin (HA) tag; MKS1, Meckel syndrome type 1; SD, standard deviation; TZ, transition zone; WT, wild-type.

necessary for the localization of Rab8 in both auditory and olfactory cilia (S6C–S6E Fig). However, we were disappointed to find that auditory cilium morphology looked normal in *rab8$^1$* mutants (S6F Fig), supported by the behavior readouts that no defects were observed in larval hearing and touch sensitivity (S6G Fig). However, we did observe that a small proportion of spermatocyte centriole pairs have elongated CG6652-GFP signal in every *rab8* mutant fly we checked (Fig 6F). Elongated CG6652 signal in spermatocytes is usually attribute to ciliary cap/TZ defects. So, Rab8 is indeed involved in ciliogenesis in *Drosophila*, even though the effect is relatively weak, probably due to the redundancy with other factors, as has been reported for Rab8 and Rab10 in mammalian ciliogenesis [42].

Multiple reports indicated that the CBY-FAM92 module facilitate ciliary membrane formation and BB docking [43,44]. Since both CBY-FAM92 module and Rab8 are involved in ciliary membrane formation, we hypothesized that these 2 DZIP1 binding partners may jointly promote early ciliary membrane formation and ciliogenesis in *Drosophila*. To test our hypothesis, we first confirmed the interaction between CBY and DZIP1 in *Drosophila* (Fig 6B and 6C). Then, we created a *cby* deletion mutants (Fig 6D), and then generated the *cby; rab8* double mutants. Interestingly, we found that the signal of the TZ component MKS1 was dramatically decreased, whereas the percentage of centrioles with abnormally elongated CG6652 signal was significantly increased in spermatocytes of *cby; rab8* double mutant, compared with either *cby* or *rab8* single mutants (Fig 6E and 6F).

Furthermore, as CBY and FAM92 function in a same pathway to regulate ciliogenesis. The TZ localization of CBY and FAM92 is interdependent, and *fam92* mutants mimic the phenotype of *cby* mutants. We wondered whether deletion of *rab8* could also aggravate the phenotype of *fam92* mutants. Then, we created *fam92* deletion mutants and generated the *fam92; rab8* double mutants. Indeed, as expected, MKS1 signal decreased dramatically, and the percentage of centrioles with abnormally elongated CG6652 signal increased significantly in spermatocytes of *fam92; rab8* double mutant (Fig 6E and 6F).

All these results strongly indicated that CBY and Rab8 do function downstream of DZIP1 and jointly contribute to TZ formation.

## Discussion

Ciliogenesis begins with the BB docking, followed by the subsequent formation of the early ciliary shaft membrane and assembly of the TZ, which together form the ciliary bud. However, how early ciliary membrane formation coordinates with TZ assembly is still not clear. In this work, we link the TZ core component CEP290 to the DZIP1-CBY/Rab8 ciliary membrane

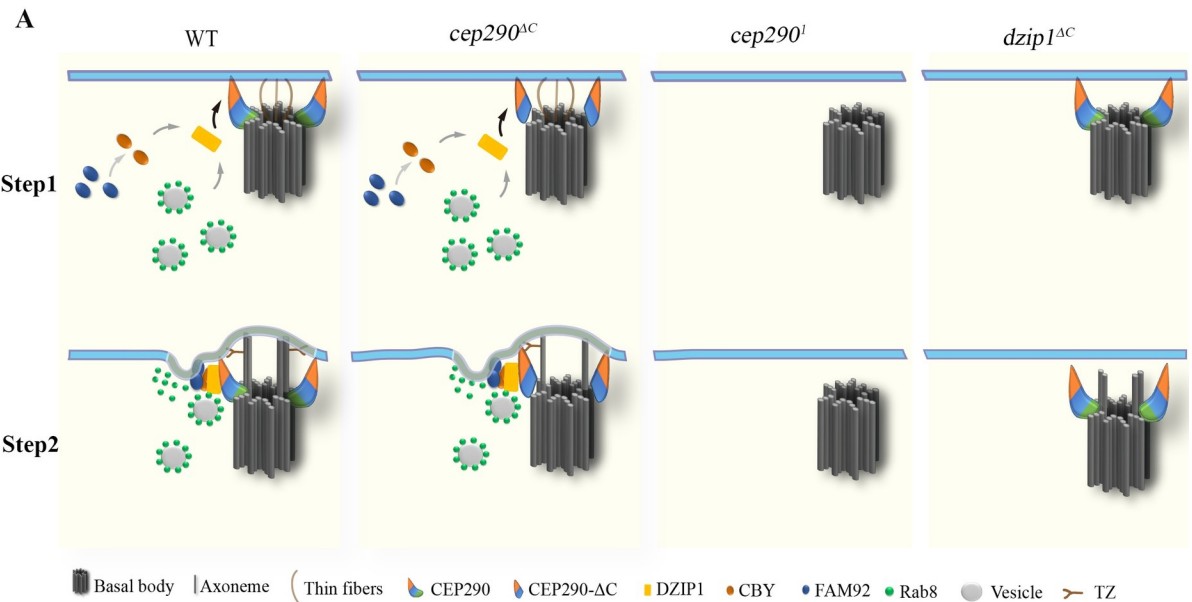

**Fig 7. Model for the initiation of TZ assembly in *Drosophila*.** Proposed model for the initiation of TZ assembly based on data from *Drosophila* spermatocytes. Thin fibers with unknown components were observed to mediate the early interaction between the BB and the PM in *Drosophila* spermatocytes [50]. CEP290 localizes at the distal end of centriole, with its C-terminus toward the centriole microtubules. After the BB migrates to the membrane, the N-terminus of CEP290 binds to the membrane and recruits DZIP1, which then recruits CBY and Rab8. CBY together with FAM92 to remodel the membrane and facilitate Rab8-mediated ciliary membrane formation, and then the TZ assembly begins. In *cep290^{ΔC}* mutant, the remaining N-terminus of CEP290 is functional; therefore, the transition zone assembly is initiated. In *cep290^1* and *dzip1* mutant, ciliogenesis is blocked at the stage of early ciliary membrane formation. BB, basal body; CBY, Chibby; CEP290, centrosomal protein 290; DZIP1, DAZ interacting zinc finger protein 1; FAM92, family with sequence similarity 92; PM, plasma membrane; TZ, transition zone.

formation module and uncover a novel function of CEP290 in coordinating early ciliary membrane formation and TZ assembly. We propose that the N-terminus of CEP290 recruits DZIP1, which then recruits Rab8 and CBY to promote early ciliary membrane formation (Fig 7), whereas the C-terminus of CEP290 associates with axonemes and regulates TZ elongation and/or maturation. Our observation that TZ assembly is completely blocked in *dzip1* mutants suggests that early ciliary membrane formation is a prerequisite for TZ assembly.

In agreement with previous reports [24,27], TZs fail to form in sensory cilia in *cep290^{ΔC}* mutants, but they still form in spermatocyte cilia, suggesting that CEP290 has a tissue-specific functional difference in TZ assembly of sensory cilia and spermatocyte cilia, 2 types of ciliated cells in *Drosophila*. This difference may be correlated with the significant reduction of DZIP1 signal in sensory cilia but not in spermatocyte cilia in *cep290^{ΔC}* mutants (Fig 5A and 5B). It has been reported that ciliary base structures are different in sensory neurons and spermatocytes [27]. Our SIM analysis of the TZ protein localization showed that there is an obvious gap, where DZIP1 is located, between TF and TZ core components in sensory neurons, but no gap exists between TF and TZ core components in spermatocytes (Fig 4D and 4E). All these structural differences may contribute to the tissue-specific functional difference of CEP290 in *Drosophila*. In addition, the TZ in *Drosophila* sensory neurons are longer than most cells and are suggested to be homologous to the specialized TZ, "connecting cilium" [45]. It is also possible that CEP290 plays a specific role in the formation of this kind of TZ. Interestingly, mammalian CEP290 may also have a specific role in the formation of "connecting cilium" of photoreceptors. In *cep290* null mice, photoreceptor cells lack cilia, while ependymal cells retain cilia [46]. In *CEP290* patients, cilia defects are often evident in photoreceptors [26,47]. Therefore, there

may be an evolutionary conservative mechanism of CEP290 in the formation of "connecting cilium" which is worthy of further investigation.

From our TEM analysis, we observed that BBs are attached/very close to the plasma membrane in *cep290*[1] mutant, similar to what Jana and colleagues has proposed [27]. However, technically, it is very difficult to determine whether BBs are docked or just apposed to the plasma membrane. In mammalian cells, the emerging model of the BB is as follows: Myo-Va–associated preciliary vesicles dock to distal appendages of the mother centriole, fuse into a large ciliary vesicle, and then the large ciliary vesicle associated with the BB migrates to and fuses with the plasma membrane [9]. However, to date, no evidence indicates that this mechanism is conserved in invertebrates. In *Drosophila*, the centriole pericentrin-like protein (PLP) regulates the migration of centriole/BB [48,49]. But how the centriole attaches to the plasma membrane remains unclear. Recently, thin fibers with unknown components were observed to mediate the early interaction between the BB and the plasma membrane in *Drosophila* spermatocytes [50]. In sensory neurons, there are *Drosophila*-specific structures called dense fibers that connect the proximal centriole (daughter centriole) to the membrane [51,52]. It is likely that those thin fibers in spermatocytes and dense fibers in sensory neurons are involved in the initial BB membrane docking. More work will be needed to understand the mechanisms of BB docking and the role of CEP290 in BB docking in *Drosophila*.

By using the simple *Drosophila* model, we demonstrated that Rab8 and CBY-FAM92 module function downstream of DZIP1 and jointly regulate TZ assembly initiation. The fact that *rab8; cby* double mutants and *rab8; fam92* double mutants show synthetic defects suggests that they have synergetic roles in ciliogenesis. It is tempting to speculate that CBY-FAM92 module promotes membrane remodeling and then facilitate the Rab8 and its yet unknown compensatory mechanism regulating the ciliary membrane formation. The data available so far suggest that the function of DZIP1-Rab8 module and DZIP1-CBY-FAM92 module in ciliogenesis is largely conserved from *Drosophila* to mammals [53]. The interaction between DZIP1 and Rab8, between DZIP1/DZIP1L and CBY, and between CBY and FAM92 are conserved [36,37,40,43], and all of them have been implicated in ciliogenesis. Similar to what we observed in *Drosophila*, *Dzip1* or *Dzip1l* knockout (KO) mice have much more severe defects than *Cby1* KO mice or *Fam92a* KO mice, supporting the notion that CBY-FAM92 is not the solely downstream pathway of DZIP1 or DZIP1L [35,36,44,54]. However, there are 2 DZIP1 orthologs (DZIP1 and DZIP1L), 2 FAM92 orthologs (FAM92A and FAM92B), and 3 CBY orthologs in vertebrates, the existence of so many paralogs makes it very complicated to study their genetic relationship.

For the first time, using the *Drosophila* model, we demonstrated that TZ core protein CEP290 directly interacts with DZIP1 and controls TZ assembly initiation by regulating DZIP1-mediated early ciliary membrane formation. It has been reported that mammalian DZIP1/DZIP1L interacts with Rab8 and CBY [36,37,40], and mammalian DZIP1L is required for ciliary bud formation and TZ assembly. Mother centrioles with docked vesicles but no ciliary buds have been observed in *Dzip1l* mutant mice [36]. Thus, the function of DZIP1/DZIP1L in early ciliary membrane formation is likely conserved in mammals [53]. Although there is no reports connecting CEP290 to DZIP1 in mammals yet, it has been suggested that mammalian CEP290 is involved in ciliary vesicle formation [55], and defects in early ciliary membrane formation have been observed in fibroblasts of *CEP290*-deficient patients [47]. In addition, our preliminary data show that human CEP290 interacts with DZIP1L (S7 Fig). Therefore, it is highly possible that the function of CEP290-DZIP1 module in TZ assembly initiation is conserved in mammals. It will be interesting to investigate and confirm our overall model for TZ assembly initiation in mammalian cells.

## Methods

### Fly stocks

The fly strains used in the study are as follows: $w^{1118}$ (FBal0018186), *elav-GAL4* (BDSC 458), *unc²* (BDSC 7424), *rab8¹* (BDSC 26173), *Df(3L)ED228* (BDSC 8086), DJ-GFP (BDSC 5417), *UASp-YFP-Rab8^{T22N}* (BDSC 9780), *UASp-YFP-Rab8^{Q67L}* (BDSC 9781), and *UASp-YFP-Rab8* (BDSC 9782); CG6652-GFP, UNC-GFP, MKS1-GFP and MKS6-GFP were provided by Dr. Bénédicte Durand; *nompC-Gal4; UAS-GFP* was provided by Dr. Wei Zhang. The flies were raised on standard media at 25˚C.

### Generation of transgenic flies

Transgenic lines were generated by the Core Facility of *Drosophila* Resources and Technology, Shanghai Institute of Biochemistry and Cell Biology (SIBCB), Chinese Academy of Sciences (CAS). To construct the FBF1-GFP, CBY-GFP, and DZIP1-GFP plasmids, the entire coding sequences (CDSs) with their promoter regions were separately cloned and inserted into the HindIII and BamHI sites of PJFRC2 vectors using a one-step cloning method. To construct the UAS-DZIP1-GFP and UAS-CEP290-GFP plasmids, we cloned the entire CDSs into the BglII and BamHI sites of PJFRC2 vectors. The DILA-GFP, truncated CEP290, and truncated CEP290-GFP plasmids were prepared similarly. We first subcloned the ubiquitin promoter region from pWUM6 into the HindIII and BglII sites of PJFRC2 vector or PJFRC2-APEX vector, and then individually cloned each gene cDNA sequence into the NotI and BamHI sites of the vector. Transgenes of CEP290, DILA, and UNC were inserted at the attP site of the 25C6 locus on chromosome 2, and transgenes of FBF1 and DZIP1 were inserted at the attP site of 75B1 locus on chromosome 3. All CDS fragments were amplified from cDNA reverse transcribed from total RNA. The primers used are listed in S1 Table.

### Generation of deletion mutants in *Drosophila*

The CRISPR/Cas9 system was used to generate the deletion mutants. Briefly, CRISPR targeting sites were designed using Target Finder (http://tools.flycrispr.molbio.wisc.edu/targetFinder). Customized single-guide RNA (gRNA) oligos were synthesized and then cloned into the BbsI site of the pEASY-PU6-BbsI-chiRNA vector. All the constructed plasmids were then injected into *vasa*-Cas9 embryos by the Core Facility of *Drosophila* Resources and Technology, SIBCB, CAS following a standard protocol. The mutant fly lines were screened via PCR. Semiquantitative reverse transcription-polymerase chain reaction (RT-PCR) was used to analyze the transcription level. The *rp49* gene was used as an internal reference. The primers used are listed in S1 Table.

### Immunofluorescence

For staining of sensory cilia or testes, pupae, or juveniles were collected 36–48 h after puparium formation (APF). The collected flies were dissected and washed with 1× PBS. The dissected tissues were then placed on slides and squashed with coverslips. The slides were submerged in liquid nitrogen for 1 min, after which coverslips were removed. After incubation in cold methanol (−20˚C) for 15 min followed by cold acetone (−20˚C) for 10 min, the slides were washed 2 times for 10 min each in 1× PBS at room temperature, and then blocked for at least 1 h in blocking buffer (0.3% Triton X-100, 3% bovine serum albumen in PBS) at room temperature and stained with primary antibodies overnight in a moisture chamber at −4˚C. After washing with 1× PBS, the slides were incubated with secondary antibodies for 3 h at room temperature.

### Cell membrane staining

Testes from juvenile flies were dissected in PBS and placed on slides. The testes were stained in CellMask Deep Red (Invitrogen, Carlsbad, CA, USA) solution (40 μg/mL) for 5 min at room temperature.

### Antibodies

Rabbit antibodies against fly DZIP1 (aa 451–737) and CEP290 (aa 292–541) were produced by YOUKE Biotech (Shanghai, China). Mouse anti-GFP (1:200, 11814460001, Roche, Basel, Switzerland), rabbit anti-GFP (1:500, ab290, Abcam), mouse anti-γ-tubulin (1:500, T5326, Sigma-Aldrich), and mouse anti-21A6 (1:200, AB528449, DSHB) were also used. The secondary antibodies (1:1000 dilution) were goat anti-mouse Alexa Fluor 488/594 or goat anti-rabbit Alexa Fluor 488/594.

### Microscopy and image analysis

Most slides were visualized using a fluorescence microscope (Nikon Ti, Tokyo, Japan) or a confocal microscope (Olympus FV1000a, Tokyo, Japan) with a $100 \times$ (1.4 NA) oil-immersion objective. SIM images were taken using a Delta Vision OMX SR (GE Healthcare, Buckinghamshire, United Kingdom) with a $60 \times$ (1.42 NA) oil-immersion objective. All data analysis was performed using ImageJ (National Institutes of Health, Bethesda, USA). For quantification of signal intensity of TZ proteins, the fluorescence intensity is calculated by subtracting the sum of background signals of the same size from the sum of the signals in the region of interest, $n$ = 50 centrioles or BBs over 5 flies.

### Transmission electron microscopy

Testis and antenna samples were prepared according to published protocols. Briefly, tissues were fixed in 2.5% glutaraldehyde buffer for at least 24 h and post fixed in $OsO_4$. The samples were dehydrated using a diluted ethanol series and embedded in epoxy resin. Ultrathin sections (approximately 70 nm) were cut using an Ultracut S ultramicrotome (Zeiss, Jena, Germany). The samples were stained with uranyl acetate and lead citrate. Electron microscopy observations were made with Hitachi H-7650 transmission electron microscope (Tokyo, Japan) at 80 kV.

### Male fertility test

Single newly enclosed males were crossed with a virgin $w^{1118}$ females for 3 days. The fertility ratio was quantified, and the progeny were counted.

### Negative geotaxis assay

For the adult climbing assay, 10 flies aged 3 to 5 d were placed in a graduated cylinder. At the beginning of each test, the flies were gently tapped to the bottom of the cylinder. The flies that climbed above the 8-cm height mark in 10 s were then counted, and counting was repeated 10 times. The data for each genotype are the averages of 5 biological replicates.

### Larval touch assay

The larval touch assay was performed as previously described [52]. A single third instar larva was gently touched on its thoracic segments with a slim eyelash during bouts of linear locomotion on an agar plate. A score was assigned according to the response of the larva: A score of 0

indicated that the larva did not respond and continued to crawl, a score of 1 indicated that the larva hesitated or immediately stopped, a score of 2 indicated that the larva anteriorly retracted with or without anterior turns, a score of 3 indicated that the larva retracted with 1 full wave of the body, and a score of 4 indicated that the larva retracted with 2 or more full waves of the body. The score for each larva was determined 5 times, and the results were summed. In each group, 10 larvae were tested, and there were 5 groups for each genotype.

## Larval hearing assay

The larval hearing assay was processed as previously described [56]. In each assay, 5 third instar larvae were placed on an agar plate. Larvae were observed for contraction responses before and during a tone of 1k Hz at 30 s intervals, repeated 5 times. Five groups were tested for each genotype. The data are presented as box plots.

## Yeast two-hybrid assay

DZIP1 was introduced into the pGBKT7 vector (Clontech, Mountain View, CA, USA), and CEP290-N (aa 1–887) and CEP290-ΔN (aa 888–1978) were individually inserted into the pGADT7 vector (Clontech). The constructs were subsequently transferred into strain AH109 (Takara Bio, Tokyo, Japan) by the LiCl-polyethylene glycol method. Yeasts were grown on SD-Leu-Trp plates at 30˚C. After 3 to 4 d of incubation, positive colonies were tested on SD-Ade-Leu-Trp-His plates and incubated for 5 d at 30˚C.

## Immunoprecipitation

To construct the DZIP1-GFP, CEP290-N-GFP (aa 1–878), DZIP1-HA, CBY-HA, and Rab8-HA plasmids, we cloned the CDSs into the *Drosophila* Gateway vector PAWG or PAWH, respectively. GFP was cloned into pAWM as a control. One microgram of total DNA was transfected into S2 cells or HEK293 cells in a 60-mm plate with Effectene (Qiagen, Duesseldorf, Germany). After 48 h, the cells were lysed in ice-cold Co-IP lysis buffer (10 mM Tris [pH 7.5], 150 mM NaCl, 0.5 mM EDTA, 0.5% NP-40, 1× protease inhibitor) at 4˚C for 45 min and centrifuged at $20,000 \times g$ for 10 min at 4˚C. The supernatants were added to anti-GFP nanobody agarose beads (Chromotek, Planegg-Martinsried, Germany), and the beads were tumbled end over end for 1 h at 4˚C. After western blotting, the polyvinylidene fluoride (PVDF) membrane was incubated with anti-GFP antibody (Roche) and anti-HA antibody (Invitrogen) or anti-Flag antibody (Sigma). Full scans of the western blots are presented in S1 Raw Image.

## GST pull-down assay

To generate the GST-DZIP1-N (aa 1–293), GST-DZIP1-C (aa 294–737), His-CEP290-N (aa 1–887), His-CEP290-ΔN (aa 888–1978), His-CBY, His-Rab8, His-Rab8$^{T22N}$, and His-Rab8$^{Q67L}$ plasmids, we cloned the CDSs into the pET28a or pGEX-4 T-1 vector. The constructs were transformed into BL21 (DE3) *E. coli*. The proteins were expressed and purified using His resin or Glutathione Sepharose. The purified His-CEP290-N, His-CEP290-ΔN, His-CBY, His-Rab8, His-Rab8$^{T22N}$, and His-Rab8$^{Q67L}$ proteins were incubated with either GST, GST-DZIP1-N, or GST-DZIP1-C in binding buffer (25 mM Tris-HCl [pH 7.4], 150 mM NaCl, 0.5% Triton X-100, 1 mM dithiothreitol, 10% glycerol and protease inhibitors) overnight at 4˚C. After western blotting, the PVDF membrane was incubated with anti-His antibodies (Invitrogen) and stained with Ponceau S. Full scans of the western blots are presented in S1 Raw Image.

## Statistics

The results are presented in all figures as the means and SDs. Statistical analyses were performed with two-tailed unpaired Student $t$ tests. (GraphPad Prism 5, Version X; La Jolla, CA, USA; ns, $P > 0.05$; *, $P \leq 0.05$; **, $P \leq 0.01$; ***, $P \leq 0.001$).

## Supporting information

**S1 Fig. Identification and phenotype analysis of *cep290* mutants.** (A) Genotyping of $cep290^1$ mutants by PCR of CEP290 fragment spanning the deletion region using whole fly genomic DNA. The amplification products were 1,754 bp long for $w^{1118}$ and 929 bp long for $cep290^1$ mutants. (B) RT-PCR analysis of the splice isoforms and transcription level of *cep290* gene in c$ep290^{\Delta C}$ mutants. Amplification of the C-terminal half of *cep290* CDS (2662–5937) did not show other splice forms caused by the deletion mutation. Semiquantitative RT-PCR analysis of the transcription level of *cep290* CDS using primer F2 and R1 showed that the transcription level was reduced in c$ep290^{\Delta C}$ mutant, probably due to nonsense-mediated mRNA decay. Housekeeping gene *rp49* was used as the internal control. (C) cDNA sequencing verification of c$ep290^{\Delta C}$ mutant. (D) Upper panel: Immunofluorescence staining with an anti-CEP290 N (aa292-541) antibody confirmed that the CEP290 N-terminal signal was reduced in $cep290^{\Delta C}$ mutants but completely lost in $cep290^1$ mutants. The corresponding relative fluorescence intensity was quantitatively displayed in lower panel. The bars and error bars represent the means and SDs, respectively. $n = 50$ centrioles over 5 flies. Scale bars, 1 μm. (E) Cilia-related behavior analysis of *cep290* mutants. Both $cep290^{\Delta C}$ and $cep290^1$ mutant flies have severe defects in hearing (left panel) and touch sensitivity (right panel). Expression of CEP290 rescued these defects. For the hearing assay, 5 larvae as a group, and at least 5 groups of flies were tested. For the touch sensitivity assay, $n = 50$. (F) Testes of WT flies and *cep290* mutants. The mitochondrial protein DJ was used to label sperm cysts. Compared to those in WT flies and $cep290^{\Delta C}$ mutants, sperm cysts in cep290$^1$ mutants were severely defective in elongation. The arrow indicates that the cysts failed to elongate. Bar, 200 μm. (G) Male fertility assay in WT flies and *cep290* mutants. We first rescued the severely uncoordinated phenotype of *cep290* mutants by expressing CEP290 driven by *elav*-GAL4. The percentage of fertile males was then quantified, and the number of offspring was calculated. $cep290^1$ males were completely infertile, while more than 65% of $cep290^{\Delta C}$ males were fertile. $n = 50$. (H) EM images of testis cross sections from WT flies and $cep290^1$ mutants. There were 64 spermatids in each cyst in WT flies, whereas axonemes were almost completely lost in $cep290^1$ mutants. The arrows show defective axonemes. (I) EM images of longitudinal sections of auditory cilia in WT flies and *cep290* mutants. Bars, 200 nm. Numerical data for panels D, E, and G can be found in the file S1 Data. CDS, coding sequence; CEP290, centrosomal protein 290; DBB, distal basal body; DJ, Donjuan; EM, electron microscope; PBB, proximal basal body; *rp49*, ribosomal protein 49; RT-PCR, reverse transcription-polymerase chain reaction; TZ, transition zone; WT, wild-type.
(TIF)

**S2 Fig. DZIP1 is the ortholog of mammalian DZIP1 and DZIP1L in *Drosophila*.** (A) DZIP1 is evolutionarily conserved in Holozoans. There is only 1 DZIP1 gene in invertebrates, while there are 2 genes in vertebrates, DZIP1 and DZIP1L, possibly because of gene duplication. (B) Phylogenetic tree of DZIP1. *Drosophila* DZIP1 is orthologous to both DZIP1 and DZIP1L. (C) Schematic representation of the protein structure of *Drosophila* DZIP1 and human DZIP1 and DZIP1L. All share highly conserved domain that includes Dzip_like-N and ZnF_C2H2. (D) Table showing the sequence similarity and identity between *Drosophila* DZIP1 and human

DZIP1 and DZIP1L.
(TIF)

**S3 Fig. CEP290–DZIP1 interaction analysis.** (A) DZIP1 did not interact with CEP290-ΔN (Δ aa 1–887) in the GST pull-down assay. (B) The GST pull-down assay was used to narrow down the interaction region between CEP290 and DZIP1. DZIP1-N (aa 1–293) interacts with both CEP290 N (aa 401–650) and CEP290 N (aa 651–887), and DZIP1-C (aa 294–737) interacts with CEP290 N (aa 651–887). Uncropped immunoblots can be found in S2 Raw Image. CEP290, centrosomal protein 290; DZIP1, DAZ interacting zinc finger protein 1; GST, glutathione S-transferase.
(TIF)

**S4 Fig. The subcellular localization of DZIP1 in *Drosophila*.** (A) Subcellular localization of DZIP1 during spermatogenesis in *Drosophila*. Upper panel: DZIP1-GFP; lower panel: anti-DZIP1. DZIP1-GFP and anti-DZIP1 staining showed similar localization patterns. No DZIP1 was present on centrioles in spermatogonia. DZIP1 began to appear on the tips of centrioles in early spermatocytes. In round spermatids, DZIP1 migrated together with the ring centriole to the tip of flagella. The centrioles/BBs were labeled with γ-Tubulin (red). Bar, 2 μm. (B) DZIP1 localized to the ciliary base in both auditory and olfactory cilia. Ciliary bases were labeled with 21A6 (red). Anti-DZIP1 staining showed a localization pattern similar to that of DZIP1-GFP. Bar, 1 μm. (C) 3D-SIM images of DZIP1 localization in spermatids, DZIP1 colocalized with CBY at the ring centriole, but did not extend like MKS1. Bar, 500 nm. (D) 3D-SIM images of spatial relationship between TF marker FBF1 and TZ core protein MKS1 in various types of cilia. Notably, there is a gap (arrows) between FBF1 and MKS in sensory cilia but not in spermatocyte cilia. Bar, 500 nm. 3D-SIM, three-dimensional structured illumination microscopy; BB, basal body; CBY, Chibby; DZIP1, DAZ interacting zinc finger protein 1; FBF1, Fas binding factor 1; GFP, green fluorescent protein; MKS, Meckel–Gruber syndrome; MKS1, Meckel syndrome type 1; TF, transition fiber; TZ, transition zone.
(TIF)

**S5 Fig. Identification and phenotype analysis of *dzip1* mutants.** (A) Generation of the *dzip1* deletion mutant. Schematic of the *dzip1* gene. The gRNA target sites are represented by scissors. *dzip1^{ΔC}* has a deletion of cDNA from nt 1,069 to 1,529, resulting in a frameshift and a premature stop codon and leading to the loss of the C-terminus. (B) Genotyping of the *dzip1* mutant. The sizes of the PCR products are as follows: 1,070 bp in *w^{1118}* flies and 609 bp in *dzip1^{ΔC}* flies. (C) Anti-DZIP1 staining confirmed that DZIP1, at least the C-terminus of DZIP1, was completely lost in *dzip1^{ΔC}* mutants. Bars, 1 μm. (D) *dzip1^{ΔC}* mutant flies showed defects in hearing and touch sensitivity, but expression of DZIP1-GFP driven by its own promoter rescued these defects. The retraction index is shown as the median and interquartile range. Numerical data can be found in the file S1 Data. The bars and error bars represent the means and SDs, respectively. *n* = 50. (E) *nompC-Gal4; UAS-GFP* was used to mark auditory cilia. Cilia completely disappeared in *dzip1^{ΔC}* mutants. (F) EM images of cross sections of cilia showing that no axonemes existed in *dzip1^{ΔC}* mutants. The black arrows indicate ciliary axonemes in WT flies. Bars, 200 nm. DZIP1, DAZ interacting zinc finger protein 1; EM, electron microscope; GFP, green fluorescent protein; gRNA, guide RNA.
(TIF)

**S6 Fig. Subcellular localization of Rab8 in sensory cilia and phenotype of *rab8* mutants.** (A) Rab8-GFP and Rab8^{Q67L}-GFP colocalized with DZIP1 at the ciliary base in auditory cilia and olfactory cilia. Notably, Rab8^{Q67L}-GFP was frequently observed in cilia. Bar, 1 μm. (B) 3D-SIM images of the colocalization of Rab8^{Q67L}-GFP and DZIP1 at the base of auditory cilia. Rab8^{Q67L}-GFP surrounded DZIP1 at TZ and expanded into cilia. Bar, 1 μm. (C) DZIP1 was

critical for Rab8-GFP localization at the ciliary base of olfactory cilia. Live imaging showed that Rab8-GFP exhibited a clear 2-dot pattern at the ciliary base in WT flies, but the signal was dispersed in *dzip1* mutants. Interestingly, these dispersed signals completely disappeared in our immunofluorescence assay, most likely due to fixation. Bars, 5 μm. (D) DZIP1 was required for Rab8 $^{Q67L}$-GFP localization at the ciliary base of auditory cilia. Bar, 1 μm. (E) Rab8-GFP was completely lost from the ciliary base of both auditory cilia and olfactory cilia in *cep290$^1$* mutants, indicating that CEP290 is essential for Rab8 localization. Bar, 1 μm. (F) Cilium morphology was normal in *rab8$^1$* mutants. (G) *rab8* mutants exhibited normal hearing responses and normal touch sensitivity. Numerical data can be found in the file S1 Data. 3D-SIM, three-dimensional structured illumination microscopy; DZIP1, DAZ interacting zinc finger protein 1; GFP, green fluorescent protein; TZ, transition zone; WT, wild-type. (TIF)

**S7 Fig. Human CEP290 N-terminal was coimmunoprecipitated with DZIP1L.** The interaction between human CEP290 and DZIP1L was analyzed using an immunoprecipitation assay. Human CEP290-N-GFP and Flag-DZIP1L were transiently transfected into HEK293 cells; 48 h later, cells were lysed and subjected to Co-IP using GFP-trap beads. Uncropped immunoblots can be found in S2 Raw Image. CEP290, centrosomal protein 290; Co-IP, coimmunoprecipitation; GFP, green fluorescent protein. (TIF)

**S1 Table. Primers were used in this paper.** (TIF)

**S1 Data. Excel file of numerical data.** (XLSX)

**S1 Raw Image. Uncropped images of immunofluorescence staining of sensory cilia.** (TIF)

**S2 Raw Image. Full scans of western blots.** (TIF)

# Acknowledgments

We thank Dr. Bénédicte Durand from Université Claude Bernard Lyon 1 and Dr. Wei Zhang from Tsinghua University for strains. We thank core facilities of *Drosophila* Resource and Technology (SIBCB, CAS), confocal imaging core facilities (SIPPE, CAS), and EM core facilities (SIPPE, CAS) for their technical support.

# Author Contributions

**Conceptualization:** Zhimao Wu, Qing Wei.

**Data curation:** Zhimao Wu, Yingying Zhang.

**Formal analysis:** Zhimao Wu, Yingying Zhang, Huicheng Chen.

**Funding acquisition:** Qing Wei.

**Investigation:** Zhimao Wu, Huicheng Chen.

**Methodology:** Zhimao Wu, Nan Pang, Yingying Zhang, Ying Peng, Jingyan Fu.

**Project administration:** Qing Wei.

**Resources:** Ying Peng, Jingyan Fu.

**Supervision:** Qing Wei.

**Validation:** Qing Wei.

**Writing – original draft:** Zhimao Wu, Qing Wei.

**Writing – review & editing:** Zhimao Wu, Ying Peng, Qing Wei.

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
