## [Editor Report · Decision Letter 0]

15 Jan 2020

Dear Dr Wei, 

Thank you for submitting your manuscript entitled "CEP290 is essential for the initiation of transition zone assembly" for consideration as a Research Article by PLOS Biology.

Your manuscript has now been evaluated by the PLOS Biology editorial staff as well as by an academic editor with relevant expertise and I am writing to let you know that we would like to send your submission out for external peer review. Nevertheless, we will need to be convinced by the reviewers that the novelty and significance of the findings is sufficient for publication in the journal. A better characterization of the role of vesicle and membrane assembly, which is a bit confusing in the study at the moment, and which other factor(s) in adition to Rab8 might be implicated in this process might be required. 

Please note that before we can send your manuscript to reviewers, we need you to complete your submission by providing the metadata that is required for full assessment. To this end, please login to Editorial Manager where you will find the paper in the 'Submissions Needing Revisions' folder on your homepage. Please click 'Revise Submission' from the Action Links and complete all additional questions in the submission questionnaire.

Please re-submit your manuscript within two working days, i.e. by Jan 17 2020 11:59PM.

Kind regards,

Ines

--

Ines Alvarez-Garcia, PhD

Senior Editor

PLOS Biology

Carlyle House, Carlyle Road

Cambridge, CB4 3DN

+44 1223–442810

---

## [Decision Letter · Decision Letter 1]

3 Mar 2020

Dear Dr Wei,

Thank you very much for submitting your manuscript "CEP290 is essential for the initiation of transition zone assembly" for consideration as a Research Article at PLOS Biology. Your manuscript has been evaluated by the PLOS Biology editors, an Academic Editor with relevant expertise, and by four independent reviewers.

The reviews of your manuscript are appended below. As you will see, the reviewers find the work very nicely done and potentially interesting, but they also raise concerns regarding the novelty of certain aspects of the findings 

They each suggests experiments to confirm and extend some of the findings, like the role of Rab8 in TZ assembly, the interaction between CEP290 and DZIP1/Rab8 or exploring thoroughtly the phenotype of the CEP290 mutants. After discussing the reviews with the academic editor, we think that in general elucidating the Rab8 axis will be necessary for us to consider the manuscript further for publication.

I regret that we cannot accept the current version of the manuscript for publication. We remain interested in your study and we would be willing to consider resubmission of a comprehensively revised version that thoroughly addresses all the reviewers' comments. We cannot make any decision about publication until we have seen the revised manuscript and your response to the reviewers' comments. Your revised manuscript would be sent for further evaluation by the reviewers.

We appreciate that these requests represent a great deal of extra work, and we are willing to relax our standard revision time to allow you six months to revise your manuscript.We expect to receive your revised manuscript within 6 months.

**IMPORTANT - SUBMITTING YOUR REVISION**

*Resubmission Checklist*

*Published Peer Review*

*PLOS Data Policy*

*Blot and Gel Data Policy*

Sincerely,

Ines

--

Ines Alvarez-Garcia, PhD

Senior Editor

PLOS Biology

Carlyle House, Carlyle Road

Cambridge, CB4 3DN

+44 1223–446970

Reviewers’ comments

Rev. 1:

In this manuscript, Wu et al. investigate the role of Cep290 in ciliogenesis and transition zone (TZ) assembly in Drosophila. By analyzing a novel null mutant of Cep290, they find that the Cep290 N-terminal domain is essential for TZ formation and for persistent membrane association of centrioles in spermatids. These defects are proposed to be due to a newly described interaction between Cep290-N and Dzip1 that is necessary for TZ localization of Dzip1. Lastly, the authors show that Dzip1 cooperates with Cby and Rab8 to promote ciliary membrane formation.

Overall, the data are of high quality and clearly presented. The significance of the findings is partly diminished by a number of studies in Drosophila and other organisms that have explored the roles of Cep290, Dzip1, and Cby in transition zone formation. However, the present work fills in important details (such as the novel connection between Cep290 and Dzip1) and carefully examines ciliary phenotypes in several Drosophila cell types. These results are thus likely to be of strong interest to the cilia field (especially if the points below are addressed). The conceptual advance is somewhat more limited, and thus the significance to a broader audience is less clear. However, based on the overall high quality of the data and the new molecular insights into transition zone formation, I believe a revised manuscript could be a good candidate for publication in PLoS Biology.

Major points:

1. While a variety of experimental approaches are employed to demonstrate binding between Cep290-N and Dzip1, a relatively large fragment of Cep290 is used (aa 1-650). This region has previously been shown to have various important functional properties (homodimerization; binding to Cp110, Pcm1, and Nphp2; membrane binding via a conserved amphipathic helix), and thus it is possible that the various phenotypic defects attributed to loss of Cep290-N are not due to loss of Dzip1 binding per se but due to disruption of one or more of these other functional elements (in fact the phenotypes seen in Cep290-l mutants that are not present in Cep290-deltaC could be due to any elements within aa 1-1385). This concern is partly mitigated by the similarity of phenotypes seen in Cep290-l and Dzip1 mutants, but it would be helpful if the authors can narrow Cep290's Dzip1 binding region more precisely and analyze ciliary phenotypes when Dzip1 binding is more selectively disrupted (e.g. by rescue of Cep290-l with such a targeted Cep290 mutant). At a minimum this issue should be discussed in more detail in the manuscript.

2. A key point in the manuscript is that the N-terminus of Cep290 has key roles in transition zone formation. This conclusion relies on the more severe phenotypes observed in the Cep290-l mutant than the Cep290-deltaC mutant. However, the predicted frameshift produced by the Cep290-deltaC allele is not experimentally verified, and the possibility of alternative splice forms and/or altered transcript stability is not considered. Given the key conclusions based on this mutant, it would be very helpful if the authors could better characterize precisely what form(s) of Cep290 are expressed in this mutant and at what levels relative to wildtype.

Minor points:

1. Is there any evidence from the authors or others indicating that the reported connection between Cep290 and Dzip1 might be conserved in other species? Similarly, can the pathogenic features of any disease-associated CEP290 mutants potentially be attributed to disruption of the Cep290-Dzip1 interaction? Experiments examining these points may be beyond the immediate scope of this paper but could make for an impactful addition; at minimum further discussion of these points would be helpful.

2. There are some confusingly worded sentences and grammatical errors throughout the text (particularly in the last section of the Results).

Rev. 2:

The transition zone is a particular structure of the ciliary base which plays many roles in cilia assembly and function. Defects in TZ assembly lead to life threatening ciliopathies. Cep290 is a core component of the TZ and has been shown to act upstream of the TZ assembly program in many organisms, but it's precise function in Drosophila is only partially investigated and how it recruits other TZ proteins is not clear.

In this manuscript, the authors describe the first Drosophila cep290 complete null mutant and show that Cep290 plays, like in many other organisms, a critical function in the initiation of ciliogenesis and TZ assembly. They demonstrate for the first time that Cep290 can be recruited to the basal body by its N- or C-terminus domains and that Cep290 directly interacts with Drosophila Dzip1. They show that Dzip1 interacts with Cby a key component of TZ assembly in Drosophila and also with Rab8. Their work also investigates if Cby and Rab8 cooperate to build the drosophila TZ.

Whereas the manuscript conclusions on Cep290 are well supported by the observations, the role of Rab8 in TZ assembly is not clearly demonstrated and the authors' conclusion is largely overstated. In particular, the phenotype of the rab8 mutant is weakly documented and does not support their conclusion. To be able to conclude on Rab8 function in TZ assembly in Drosophila, the authors should at least address the following points:

-show the rescue of the phenotype of cby; rab8 double mutant by Rab8-GFP and, even more demonstrating, not by Rab8T22N.

-the sole phenotype described for the cby; rab8 double mutant is a reduction in MKS1-GFP at the TZ? Is this the only phenotype observed for this double mutant? Is there any consequence on behavioral assays? What is the testis phenotype of the rab8 mutant alone and the double cby; rab8 mutant? Is it consistent with a function in TZ assembly? My opinion is that the results on rab8 are much too preliminary to support a role of Rab8 in TZ assembly and actually reduce the impact of the other conclusions that are very interesting.

The other following points also need to be addressed before publication:

Major points :

-The authors nicely show that basal bodies (BB) in cep290 mutants do not build any TZ and are, at best, just apposed to the plasma membrane. This result was already proposed by Jana et al., 2018 using RNAi. However, it is not clear why the authors' conclude that the basal bodies are docked: in particular do the authors observe a thickening of the plasma membrane and connections between the membrane and the BB, as described by Gottardo et al, 2018? How many centrioles were observed by EM? Are they all apposed to the plasma membrane ? On Figure 2G : one of the two basal body does not seem to be docked to the plasma membrane: can this be explained by the level of the section? Do the authors have other sections of this same centriole?

-The authors analyze the distribution of different Cep290 truncated fragments fused to GFP. This analysis is apparently done in WT conditions where the endogenous protein is present, what happens in cep2901 rescue conditions ? Could the difference in the distribution of Cep290 truncated fragments just reveal differences in competition behavior between the truncated and Wt protein?

Minor points :

Overall the IF images of the sensory cilia are of poor resolution. In particular the images of sensory neurons in 1B, 1C, 4F, 5B and D.

Figure 1. The authors only show one example of the distribution of the different Cep290-GFP fragments: is the distribution homogenous between centrioles? Please quantify the length on several centrioles to show the distribution.

Figure 2C : Quantify the fluorescence intensity in cep290deltaC mutant compared to WT: is there any effect of deleting only the C-terminus, as the flies are hypofertile, and taking into account that mecH alleles show defects in TZ integrity in the testes (Basiri et al., 2014).

Figure 4D: it was previously shown in Jana et al, 2018, that Cep290 radially extends from the center to the periphery of the TZ, depending on whether GFP is fused to the N-ter or C-ter of the protein. This should be taken into account in the interpretation of the high resolution imaging observations.

Page 6: the authors propose that expressing the N-ter-Cep290 fragment has a dominant negative effect : are all transgenes inserted on the same platform as this is not explained in the methods section? Thus, in all the experiments based on the expression of these truncated forms, could there be an effect of the expression levels?

Page 7: The authors propose that the C-ter and N-ter domains have an inhibitory effect on their localization and cite Drivas et al. to support their hypothesis. However this latter article shows that overexpressing the N or C-ter domain of human CEP290 leads to increased ciliogenesis in cells. This is completely different to what the authors observe here in their manuscript and is misleading.

Figure 7 and discussion: there is an apparent confusion in the text and in the figure between what the authors call dense fibers on page 20, which in the referred publications correspond to lateral connections of the proximal (daughter) centriole and the dendrite membrane and what is called "thin fibers" described in Gottardo et al. 2018. These are completely different structures that likely have completely different roles and Figure 7 must be corrected to "thin fibers" as described in Gottardo et al. The EM images of this manuscript do not show that these connections are still present in cep290 mutants. In addition, to my knowledge, ciliary vesicles have not been described on Drosophila EM images, thus parts of Figure 7 are speculative.

At many occurrences, the conclusions appear to be overstated:

Page 8, end of second paragraph :

« Probably its N-terminus alone » this is likely overstated. It shows that the N-terminus is required but not sufficient as the C-terminus truncated Cep290mecH mutants do have a TZ gating defects (Basiri et al., 2014).

Page 13: "DZIP1 did not affect the initial docking of the BB": same comment as above for the cep290 mutant. What are the arguments to say that the BB are docked and not only apposed to the plasma membrane?

Page 20: second paragraph: "suggests that they have synergistic roles in early ciliary membrane formation". The manuscript only shows that MKS1 is affected not that there are defects in early ciliary membrane formation.

Last paragraph of the result section: the last sentence is highly speculative and should be removed.

Some of the results obtained for dzip1 mutant confirm recently published results by Lapart et al, 2019. This should be stated in the result or discussion section.

As well, some observations of the cep290 mutants confirm previous published results by Basiri et al., Jana et al. and Lapart et al., and this should be recalled accordingly in the text.

Page 21 "regulating DZIP1 mediated early ciliary membrane": incomplete sentence.

Rev. 3:

In this manuscript, Wu et al. report that the N-terminal region of CEP290 plays an essential role in the initiation of transition zone assembly of the cilia in in neurons and spermatocyte of Drosophila. They demonstrate that the N-terminus of CEP290 directly interacts with DZIP1 and recruits DZIP1 to transition zone, where it recruits CBY and RAB8 to assemble the ciliary membrane. The authors propose that CEP290 coordinates the early ciliary membrane formation and transition zone assembly.

Overall, this work is of importance to the ciliogenesis field because little is known about the regulation of early ciliary assembly events. Requirements for CEP290 in transition zone assembly and function has been investigated in several model systems including Drosophila (Jana et al 2018, Li et al 2016). Importantly, CEP290 has previosuly been reported to act upstream of DZIP1 recruitment - which was not mentioned in this paper. Moreover, associations with Rab8 function have been reported and were for the most part noted in this manuscript with exception a paper in 2008 by Kim et al. Thus, the main findings from this work largely relate to the identification of interactions for CEP290-DZIP1 and DZIP1-CBY/Rab8. However, the interaction between DZIP1 and Rab8 and CBY seem rather weak compared to the input. Another concern I have is that conclusions drawn about TZ structure and protein localization may be over interpreted from the provided SIM imaging. The quality/resolution of these images seems lower than what has been published by others for the TZ in Drosophila.

Additional concerns/questions/suggestions are listed below.

1. The C-terminus of CEP290 has microtubule binding domain while the N-terminus has membrane binding motif. Here the authors show that the N-terminus alone localizes to the transition zone (Fig. 1). How might this fragment be anchored at the transition zone and basal body? Understanding this question could help enhance the mechanistic understanding of this work.

2. The 21A6 antibody recognizes two domains longitudinally in auditory neurons. Which domain is shown in Fig. 1B, 4F and 5B? What are the directions of the cilia in these figures? Showing un-cropped images may help to predict the position of the cilia in these cells.

3. What are the stages of the spermatocyte in Fig 2C? What structure is the elongated UNC signal related to when the ciliary bud formation is blocked in CEP290 null cells as shown in Fig. 2C and 2F?

4. The localization of CG6652 in cep290 null cells is very different comparing to wt and delta C cells in Fig. 2D. The authors claim that this is abnormally extended axoneme. But the distribution of CG6653 does not look like an axoneme. Electron microscopy analysis would be necessary to confirm this structure.

5. The association between basal bodies and plasma membrane was not defective in later spermatocytes and spermatids in CEP290 dC mutant cells (Fig. 2E). The transition zone can be assembled too (Fig. 2C). Are the cilia buds formed in these cells comparing to wt and null cells as in Fig. 2F and 2G?

6. The authors indicate that DZIP1 occupied the gap region between FBF1 and MKS1 in Fig. 4E and S3E. However, there is still a gap between DZIP1 and FBF1 in Fig. 4D auditory neurons. Are these representative images?

7. It would be helpful to note in the figure legends that some of the blots are antibody staining while other are are Poncuau S. stained for clarity.

8. Showing images of the whole auditory and olfactory systems can help confirm signal from background. The cropped area in images in Fig 1BCD, Fig.3C, Fig. 4F and Fig. 5AB are too small.

Rev. 4:

The manuscript Wu et al. is a very carefully crafted, well executed and powerful genetic analysis of the role of well-known transition zone factor CEP290 in assembly of different types of cilia in Drosophila. Coupling powerful fly genetics, with rapid transgenics and reporters and quantitative imaging, the authors have created an elegant study into how CEP290, a key human ciliopathy gene, is mechanistically driving assembly of the transition zone in these highly modified cilia types. Indeed, understanding distinct and overlapping properties of the transition zone of different cilia types is key towards understanding the clinical pleiotropy observed in human ciliopathy patients in terms of symptom severity and penetrance between different tissue types. As such the study by Wu et al. is timely and of general interest to an audience like that of PLoS Biology. The tangible and highly subjective aspect of novelty of this study has been recently dinged by the Lampard et al in eLife (December 2019) from the Durand lab, where they looked at the same problem in the same system from a slightly different perspective, focusing on DZIP1 and FAM92, as well as CBY and of least focus CEP290. I believe the studies are complementary, the findings by Wu et al supporting and extending on those in the Lampard study- there is something important to be said for reproduction/confirmation of findings. However, given this and in light of the ethos of PLoS Biology to publish 'outstanding scientific significance', I feel this is somewhat an editorial issue for PLoS to decide where this one falls. Indeed, journals like eLIFE have a clear policy on publishing 'scooped' results of high quality and scientific significance, even if the novelty has been compromised. My gut feeling is it is better placed there.

Scientifically I can comment- as shown below. Generally, it is a well-written paper, although many of the figures are too busy/too many panels such that details cannot be seen but this is not fatal. It is an excellent example of genetics, imaging and biochemistry to carefully phenotypically and structurally dissect cellular process- I think it is very elegant in its execution and very thorough, very strong in its analysis. There is a propensity in the discussion to overextend speculation in the mammalian system for some of the points, but I will address them below.

Major points:

1. Accessibility: Many of the figures are too complex- it is very difficult to see details/labels even at their current full page size (see all Figure 2, Figure 3C graphs, Fig4D,E graphs, Fig 7). Also as most of the panels have only two dyes/fluorophores- I would strongly urge the authors to consider a color blind palette (magenta/green, cyan/orange- see - https://imagej.nih.gov/ij/docs/guide/146-9.html- about 10% of your male audience may be colorblind). Also the dark blue/indigo of the CellMask in 2E, 4H-I and 5F is difficult to discern from the black background- consider magenta or something bright.

2. Allelism: This current study does not address any about protein levels between lines- but puts it down to differences in activities of different protein domains- which should remain an interpretation of the results, given it could be down to just protein levels. There is a big point made about the previously published fly line (Basiri et al 2014; cep290mecH) which causes a premature termination codon resulting from a frameshift, in the 5th last exon of a very large gene- not being a null allele and instead being a milder loss-of-function allele due to remaining action of the C-terminally truncated protein (lacking the final~600 aa). Showing my mammalian bias, one would think this would subject the transcript to nonsense mediated decay, instead of protein truncation. Although speculated this truncated protein species exists, it is never investigated. I know that antibodies are difficult for Drosophila studies but some RNA/semi-quantitative PCR to look at levels and altered splice variants could lead credence to this- and that the cep290l and cep290�C lines generated here by genome editing, where larger on-target 'off-target' effects like inversions or larger deletions, make a phenotype seem more severe. Similarly, the transgenic over-expression some of the GFP CEP290 variants result in defective motility where the N-terminal CEP290 'may play a dominant negative role'- if this was true then heterozygous carriers for the cep290�C allele would have a similar uncoordinated, motor phenotype which they do not. Indeed according to the Basiri paper 'cep290mecH homozygotes (cep290) and hemizygotes (cep290/−) cannot stand….cep290mecH heterozygotes (cep290/+) and rescue (Cep290 and Cep290-GFP) are normal.' The lines of argument here are contorted. A safer approach would be like the Lampard paper uses an enhancer trap line- and just says it is a strong hypomorph and has a similar phenotype- loss of FAM92 and DZIP from the transition complexes, and '76% of aberrant axonemal growth, suggesting that basal body to membrane attachment is compromised in this mutant'. Genetics is complicated and sometime a rabbit hole not worth completely going down to prove your alleles are in fact nulls. Indeed in the clinical setting, we know that generally the severity of CEP290-associated phenotypes seems to correlate with the residual amount of functional protein that a mutant allele could produce (Drivas et al., 2015; Shimada et al., 2017)- something you have not looked at here rather tie it to domains. This is a weakness for me.

Minor points:

1. Awkward and inexact: Abstract 'Cilia play critical roles in organ generation and maintenance' consider 'during embryonic development and adult homeostasis'.

2. Introduction, p3, first line- typo '…that extend from the surface of most eukaryotic…'.

3. Introduction, p4, second line- typo '…which connect proximal axonemal…'.

4. Introduction, p4, second line Long sentence consider splitting- 'to the ciliary case membrane. These have been proposed to organize… on the membrane surface. The transition zone functions…'.

5. Introduction, p4. Second paragraph, third line. Italicize gene 'CEP290' and consider rewording from list of disorders to a more informative statement for non-expert readers like '…several ciliopathies ranging from non-syndromic retinal degeneration (i.e. Leber Congenital Amaurosis (LCA) to syndromic disorders including Senior-Loken syndrome (SLS),…'

6. Introduction, p4. Second paragraph, missing text. '…diseases and phenotypes associated with CEP290 mutations highlight the multiple and critical…'. Gene names should always be italics.

7. Introduction, p.5- inconsistency. I may be missing something but the statement 'We proposed that ciliary membrane assembly is a prerequisite for TZ assembly initiation' seems to be not what it is shown in your summary figure 7- where binding of CEP290 to apical membrane and recruitment of TZ facilitators DZIP and CBY occur before recruitment of Rab8/ciliary membrane vesicles. To the model is the inverse of this 'We proposed that TZ assembly initiation is a prerequisite for ciliary membrane assembly…'

8. Results, p.5, last paragraph, missing text. '…microtubules and ciliary membranes, where its C-terminus…'. Consider referring to Figure 1A here.

9. Results, p.5/6- clarity. Consider emphasizing constructs are overexpressed in wild type flies. Consider removing dominant negative statement at end of paragraph (see above).

10. Figure 1 A-D- do you know that all these GFP constructs are expressed- the lack of staining at the basal body of cilia for CEP290-M could simply be due to unstable protein. Also- I am perplexed: Figure 1B vs 1E- why such a different look to the cilia localizations between modalities- the confocal has two closely aligned structures and the SIM data only has one a field of view that is only double the magnification of the first, according to scale bars. Maybe a statement clarifying this is needed in the legends.

11. Results, p.7, for transparence- add reference to previously published alleles for cep290. 'Similar to previously reported alleles, both cep2901 and cep290�C were uncoordinated….. (Basiri et al 2014, Lampard et al 2019).

12. Figure 2A- for clarity, consider annotating where cep290mecH falls. Also explicitly add into the figure legends that these are predicted protein sizes- both of your CRIPSR alleles (and cep290mecH) may be subject to NMD and thus have little protein of any size to speak of.

13. Results, p.9, last paragraph- clarify. '…leading to loss of cilia/TZ' as the paragraph is written these structures never form- consider 'completely blocked, such that cilia with transition zones never form'. The following sentence may need some gentle reworking for non-fly enthusiasts, '…resulting in subsequent abnormal elongation of membraneless ciliary axonemes within the cytoplasm'.

14. Results, p.10, typo, first sentence. '…the initiation of TZ assembly, spermatogenesis….'.

15. Results, p.11, typo, second sentence. '…intensities were severely reduced compared to….'.

16. Results, p.11, emphasis. Tone down final statement first paragraph- ' …in sensory neurons, the CEP290 N-terminus may be essential for initial TZ assembly, whereas TZ elongation or maturation may involve a tissue-specific requirement for the C-terminus of CEP290.'

17. Results, p.11, typo. '… we employed a yeast two hybrid….' Also- 'Drosophila DZIP1 is the ortholog of mammalian DZIP1..'

18. Results, p.11, clarification. What is the CEP290-�N construct? What is in it is never detailed anywhere and is confusing with what is outlined in figure 1A.

19. Results, p.12, clarification. 'While the role of Drosophila DZIP1 in ciliogenesis was unknown when we started this project, a very recent publication has confirmed our findings (Lampard et al 2019).'

20. Results, p.13, typo. '…completely lost but CEP290 persisted…'. Also 'Furthermore, similar to cep2901 mutants…'

21. Results, p.14, typo. '… C-terminus may have tissue-specific roles in regulating…'.

22. Results, p.17, typo. '…compensate for the loss of Rab8, as has been reported for Rab8 and Rab10 in mammalian ciliogenesis…'.

23. Results, p.17, clarify. I was confused by the statement '…then facilitate the Rab8- and its redundant player- mediated…'. Consider '…then facilitate the Rab8- and its yet-unknown compensatory mechanism regulating…'.

24. Discussion, p.18, see point 7. '…TZ assembly is completely blocked in dzip1 mutants suggests that early ciliary membrane formation is a prerequisite for TZ assembly initiation'. Again I think this is the opposite of what your data shows- TZ assembly predates early ciliary membrane formation in your analysis.

25. Discussion, p.18, careful. The discussion about CEP290 mutations in human patients- there are retinal specific isoforms for many key RP genes which are believed to be the reason for the syndromic versus non-syndromic divide for some of these clinical phenotypes. This should be referenced. Again although the genotype-phenotype correlation is challenging, in general it is believed that the severity of CEP290-associated clinical phenotypes appears to reflect the residual amount of functional protein produced by a mutant allele (Drivas et al., 2015; Shimada et al., 2017). This current study does not address any of this- but puts it down to protein domains- which should remain an interpretation of the results, given it could be down to just protein levels.

26. Discussion, p.19, typo. 'In Drosophila, the centriole…'. Also take out the '(our unpublished data)' , either you show it or don't mention it at all.

27. Discussion, p.20, typo. '…understand the mechanisms of…'.

28. Discussion, p.20, clarify (see point 23). Change '…then facilitate the Rab8- and its redundant player- mediated…'. Consider '…then facilitate the Rab8- and its yet-unknown compensatory mechanism regulating…'.

29. Discussion, p.20, nomenclature. Mouse alleles should be title case italics- Dzip1 or Dzip1l. See final paragraph too. There is also a typo here- mice is plural so should be 'have much more severe…'. Same for Cby and Fam92a. Within species, they are not orthologs, they are paralogs. Final sentence in paragraph change to '…the existence of so many paralogs complicates the study of their genetic relationship'.

30. Discussion, p.21, missing word. '….regulating DZIP1-mediated early ciliary membrane WHAT?….'.

31. Methods, p.24, add information. The microscopy and image analysis- how was the quantitative imaging done- how many flies, how many basal bodies/TZ to get traces'.

32. Acknowledgements, p.35, typo. Change to '…for their technical support'.

33. Figure Legends, 4 title, clarify. '…with the N-terminus of CEP290, and dzip1�C deletion mutants…'

34. Supplementary Figure legends, p.42. Clarify PCR is of what- genomic DNA? In figure S1E- there is a typo throughout with your driver- should be elav-GAL4 not ealv. Figure S1- add in what acronym DBB and PBB mean.

35. Supplementary Figure legend 2, p.43. There is a typo in the figure for H. Sapiens several times.

36. Supplementary Figure legend 3, p.44. Typo '…to the tip of the flagella.'

---

## [Decision Letter · Decision Letter 2]

18 Nov 2020

Dear Dr Wei,

Thank you for submitting your revised Research Article entitled "CEP290 is essential for the initiation of transition zone assembly" for publication in PLOS Biology. I have now obtained advice from the original reviewers and have discussed their comments with the Academic Editor. 

Based on the reviews (attached below), we will probably accept this manuscript for publication, assuming that you will modify the manuscript to address the remaining points raised by the reviewers. We would also like you to consider some suggestions to improve the title. Please consider one of these two alternatives:

"CEP290 is essential for the initiation of ciliary transition zone assembly" or

"CEP290 is essential for the initiation of transition zone assembly in cilia"

Please also make sure to address the data and other policy-related requests noted at the end of this email.

We expect to receive your revised manuscript within two weeks. Your revisions should address the specific points made by each reviewer. In addition to the remaining revisions and before we will be able to formally accept your manuscript and consider it "in press", we also need to ensure that your article conforms to our guidelines. A member of our team will be in touch shortly with a set of requests. As we can't proceed until these requirements are met, your swift response will help prevent delays to publication.

- a cover letter that should detail your responses to any editorial requests, if applicable

*Copyediting*

*Published Peer Review History*

*Early Version*

Sincerely,

Ines

--

Ines Alvarez-Garcia, PhD

Senior Editor

PLOS Biology

Fig. 1E, F; Fig. 2C; Fig. 3C; Fig. 4D, E; Fig. 6E; Fig. S1D, E, G; Fig. S5D and Fig. S6G

Reviewers’ comments

Rev. 1:

In this revised manuscript, Wu et al. largely address the concerns I raised previously. While it is unfortunate that the Cep290-Dzip1l interaction could not be dissected more precisely, the authors made a reasonable effort, and they revised the text accordingly. I would encourage the authors to include the coIP showing interaction of mammalian Dzip1l and Cep290 if possible. There are also some grammatical errors that should be corrected before publication, and parts of the Discussion are not as clearly written as the rest of the manuscript. These minor issues notwithstanding, I believe the manuscript is suitable for publication in PLoS Biology.

Rev. 2:

The authors have answered to all my initial requests and the additional data reinforce their conclusions.

In particular they have corrected all the overstatements and added interesting data showing functional interactions between Fam92 and Rab8, indicating that Rab8 and Fam92 or Cby cooperate during spermatogenesis.

Minor point:

Page 13 fourth line: MKS6 is not presented on sup Fig 4C.

page 14 last paragraph: "is still existed" check English grammar.

Rev. 3:

The revised manuscript has been improved and addressed most of my concerns. Though how the N-terminus of CEP290 is anchored to the basal body to recruit Dzip1 is not completely clear. Overall this work and model proposed for CEP290-Dzip1-Rab8 pathway in TZ assembly initiation is important to the field.

Rev. 4:

The revised work by Wu et al represents a significant genetic and biochemical study into early stages of ciliogenesis in Drosophila regulated by CEP290 through the DZIP1-CBY/Rab8 module. Using a range of imaging modalities and generating many genetic tools, the work is novel and sound- appropriate for publication in a revised manner in PLoS Biology.

1. Accessibility: I am disappointed the authors did not choose to make their data more accessible to the broad readership of the journal. 10% of all readers, including some of the corresponding authors whose work they cite here, are red-green colorblind and as such are unable to interpret the bulk of the data presented in this manuscript. Changing pseudocolors in two color panels is very easy in FIJI with plug-ins- it would take the authors a half-day tops but would be so inclusive, improving pick up and citation of their work. (On a side note, I do wish PLoS would make this policy and mandate, not be optional).

2. Allelism and expression: as I wrote, we know from human genetics that levels of variant expression are key for CEP290 phenotypes- this is ultimately at the protein level, which the authors are unable to look at. I think this statement needs to be made explicitly as a caveat to their interpretation in the discussion. Something akin to 'Without a robust antibody to survey expression to validate levels of CEP290 protein expression within our alleles, we cannot fully unpick how much of our phenotypes are due solely to reduced levels versus requirement for specific domains. We suspect it is likely a combination of both'. Reviewer 1 (Question 2) also raises issues of protein levels, as does reviewer 2 who wrote '…all the experiments based on the expression of these truncated forms, could there be an effect of the expression levels?' Their response 'To ensure that CEP290 and its truncations were expressed at a comparable level, all these transgenes were inserted o the same integration platform (the attP site of 25C6 locus on chromosome 2), and they were driven by the same ubiquitin promoter. Therefore, it is unlikely that the different defective phenotypes with expressing various CEP290 truncations are due to their different expression levels'. Again, this reflect only transcript level and not stability of the protein- this is emphasized in Supp Fig 8 where there is no GFP expressed for CEP290-M suggesting it is not stable. Without a GFP blot to show similar levels of protein expression, untangling protein instability/levels versus changes in localization cannot be unpicked.

Minor points:

1. A comment: My point about was about on-target 'off-targets' in their crispant alleles- larger or complex rearrangements at the locus that because they are at the locus and cannot be 'eliminated by outcrossing 3 times to dilute any potential off targeted lesion'. Large long read sequencing is necessary and/or southern blots to really confirm, but I suspect were not done.

2. Last paragraph of the introduction- it is all past tense, but work you describe in the paper- present tense is more apropos.

3. Figure 1C legend: Sequencing of what gDNA or cDNA? Also in this figure why are the numbers different at the bottom of the chromatograms- consider removing as they are confusing, without adding value.

4. p7 'This is consistent with the transcription level of cep290 in cep290ΔC mutant' should read ' consistent with the transcript level'- you are assaying only transcript level (result of production and turnover).

5. P8 Protein DILA should be all caps.

6. Top of p12 typo : 'Drosophila DIZP1 is' should read 'Drosophila DZIP1 is'

7. Page 12 Overexpression in a cell line does not constitute in vivo 'GFP-tagged CEP290 and HA-tagged DZIP1 in cultured Drosophila S2 cells'- removing 'also occurs in vivo,' without losing emphasis.

8. Top of p16- typo 'cby or fam92 single mutants'

9. Top of p18- typo single or plural- clarify 'Then we created a fam92 deletion mutants'

---

## [Editor Report · Decision Letter 3]

16 Dec 2020

Dear Dr. Wei,

I am writing concerning your manuscript submitted to PLOS Biology, entitled “CEP290 is essential for the initiation of ciliary transition zone assembly.”

We have now completed our final technical checks and have approved your submission for publication. You will shortly receive a letter of formal acceptance from the editor.

Kind regards,

PLOS Biology